# A Decade of Progress in Deep Brain Stimulation of the Subcallosal Cingulate for the Treatment of Depression

**DOI:** 10.3390/jcm9103260

**Published:** 2020-10-12

**Authors:** Sharafuddin Khairuddin, Fung Yin Ngo, Wei Ling Lim, Luca Aquili, Naveed Ahmed Khan, Man-Lung Fung, Ying-Shing Chan, Yasin Temel, Lee Wei Lim

**Affiliations:** 1Neuromodulation Laboratory, School of Biomedical Sciences, Li Ka Shing Faculty of Medicine, The University of Hong Kong, L4 Laboratory Block, 21 Sassoon Road, Hong Kong, China; sharaf@hku.hk (S.K.); fyngo1@connect.hku.hk (F.Y.N.); weilingl@sunway.edu.my (W.L.L.); fungml@hku.hk (M.-L.F.); yschan@hku.hk (Y.-S.C.); 2Department of Biological Sciences, School of Science and Technology, Sunway University, Bandar Sunway 47500, Malaysia; 3School of Psychological and Clinical Sciences, Charles Darwin University, NT0815 Darwin, Australia; luca.aquili@cdu.edu.au; 4Department of Biology, Chemistry and Environmental Sciences, College of Arts and Sciences, American University of Sharjah, Sharjah 26666, UAE; naveed5438@gmail.com; 5Departments of Neuroscience and Neurosurgery, Maastricht University, 6229ER Maastricht, The Netherlands; y.temel@maastrichtuniversity.nl

**Keywords:** deep brain stimulation, treatment-resistant depression, major depressive disorder, subcallosal cingulate, medial prefrontal cortex

## Abstract

Major depression contributes significantly to the global disability burden. Since the first clinical study of deep brain stimulation (DBS), over 446 patients with depression have now undergone this neuromodulation therapy, and 29 animal studies have investigated the efficacy of subgenual cingulate DBS for depression. In this review, we aim to provide a comprehensive overview of the progress of DBS of the subcallosal cingulate in humans and the medial prefrontal cortex, its rodent homolog. For preclinical animal studies, we discuss the various antidepressant-like behaviors induced by medial prefrontal cortex DBS and examine the possible mechanisms including neuroplasticity-dependent/independent cellular and molecular changes. Interestingly, the response rate of subcallosal cingulate Deep brain stimulation marks a milestone in the treatment of depression. DBS achieved response and remission rates of 64–76% and 37–63%, respectively, from clinical studies monitoring patients from 6–24 months. Although some studies showed its stimulation efficacy was limited, it still holds great promise as a therapy for patients with treatment-resistant depression. Overall, further research is still needed, including more credible clinical research, preclinical mechanistic studies, precise selection of patients, and customized electrical stimulation paradigms.

## 1. Introduction

Major depressive disorder (MDD) contributes significantly to the global disability burden and social burden [1,2]. In the US from 2005 to 2010, the economic burden of patients with major depressive disorder increased by 21.5% to $210.5 billion [3]. The main symptoms of MDD include severe sadness, anxiety, cognitive deterioration, and suicidal thoughts [4]. Although its etiology is uncertain, genetic predisposition, developmental deficits, hormonal imbalance, and a stressful lifestyle may increase the risk for MDD [5,6,7,8,9,10].

Prior to the discovery of antidepressant medication, surgical ablation was used to effectively treat MDD in the US and Europe [11]. Pharmacological antidepressants first appeared in the late 20th century and these first-generation drugs became the first line treatment for depression [12]. However, newer generations of antidepressants were barely more effective than first-generation tricyclic antidepressants [13] and this has led to the emergence of treatment resistance. Treatment-resistant depression (TRD) is the failure to respond to the three different classes of treatment: antidepressants, psychotherapy, or electroconvulsive therapy given at a sufficient dose and time [14,15]. Approximately 20% to 30% of patients are refractory to pharmacotherapy and nearly 60% respond inadequately [16,17], which can result in worse clinical responses, leading to additional social burdens [18]. As the pathogenesis of MDD involves multiple structures, a broad-acting safe therapy needs to be developed [19,20].

With much progress in surgical techniques and advances in cardiac pacemakers, electrical stimulation has matured to become an adjustable stimulatory regimen [21]. Deep brain stimulation (DBS) is a procedure whereby deep brain structures are stimulated via precisely implanted electrodes. It was first used to alleviate movement disorders in patients with Parkinson’s disease [22]. With advances in our understanding of the limbic circuitry, the focus has shifted to the antidepressant-like effects of DBS [23]. Some recent clinical studies have shown that DBS holds great promise in treating patients with TRD, and mechanistic studies in animals are currently in progress.

The use of DBS as a treatment for TRD was first proposed in a study by Kruger et al. on the differences in regional cerebral blood flow (rCBF) between remitted patients and bipolar depression (BD) patients [24]. They observed that rCBF in Brodmann Area 25 (BA25) was higher in remitted and BD patients compared to control patients and this was also seen in healthy patients with self-rated high negative affect [25]. Furthermore, Kruger et al. noted that mood provocation did not change the rCBF to this region in BD patients compared to MDD patients, indicating that dysfunction in the region was specific to depression [24]. Mayberg et al., who are pioneers of DBS as a treatment for depression, subsequently targeted BA25 after detecting metabolic abnormalities within the region that were consistent with those found in patients with TRD [19]. This landmark paper led to further developments in the application of DBS of this region as a treatment for depression.

Indeed, several research groups have used DBS to treat depression by targeting different brain regions in the limbic system. Jimenez et al. applied DBS to the inferior thalamic peduncle, whereas Schlaepfer et al. applied DBS in the nucleus accumbens core [26,27] and successfully performed DBS on the medial forebrain bundle [28]. With rapid developments in DBS as a treatment for TRD, research is now focusing on the subcallosal cingulate (SCC). This review aims to examine and consolidate clinical and preclinical research on the use of DBS as a treatment for depression, targeting the subcallosal cingulate in humans and the ventromedial prefrontal cortex, the anatomical correlate in rodents.

## 2. Outline of the Review

The online PubMed database was searched for research articles in English using a Boolean operation with keywords including “deep brain stimulation” AND “depression” AND “subcallosal cingulate” OR “rodent” AND “medial prefrontal cortex”. Relevant articles cited in the reference lists of the identified publications were also included. PubMed was utilized due to its extensive collection of indexed peer-reviewed journals. This review highlights the development of DBS as a treatment for TRD and discusses the findings and limitations of preclinical and clinical studies published in the recent decade. The neuroplasticity-dependent and -independent aspects of the molecular and cellular changes due to DBS are also discussed. Lastly, some potential approaches that may improve the precision, safety, and efficacy of DBS are proposed.

## 3. The Development of Deep Brain Stimulation as a Treatment for Depression

Deep brain stimulation involves the stereotactic implantation of thin electrodes in deep brain structures that are used to deliver electrical stimulation generated by a subcutaneous pulse generator [29,30]. Stimulation is generally applied at either a low/moderate frequency (5–90 Hz) or high frequency (100–400 Hz). Since the inception of DBS, a number of studies have demonstrated that this modality has the ability to treat pain, obsessive-compulsive disorder, and Parkinson’s disease [16,21,31]. Its efficacy has been verified in Parkinson’s disease patients, in which high frequency stimulation (HFS) of specific brain region(s) in the basal ganglia was able to stop tremors [21,32]. The use of DBS has been given FDA approval for the management of obsessive-compulsive disorder since 2007, but it is only provided under a humanitarian device exemption [33,34].

The following sections summarize the clinical studies on deep brain stimulation in the subcallosal cingulate for treating patients with treatment-resistant depression and preclinical studies of deep brain stimulation in the medial prefrontal cortex (mPFC) of rodents.

## 4. Clinical and Preclinical Studies of SCC DBS for the Treatment of Depression

Clinical studies of depression utilize rating scales of depression that assess changes in depressive symptoms in patients. Some scales are completed by the researcher such as the Hamilton Depression Rating Scale (HDRS) and the Montgomery-Åsberg Depression Rating Scale (MADRS). These rating scales should allow more consistent assessment between patients, but can lack consensus in their interpretation among researchers, which could lead to misdiagnosis [35]. Another weakness of rating scales conducted in this manner is that the accuracy of the results is dependent on the communication skills of the patient, which might be hampered by the disease itself. Other scales are completed by patients such as the Beck Anxiety Inventory (BAI), Beck Depression Inventory (BDI), and Quick Inventory of Depressive Symptomatology (QIDS). These rating scales should allow for a more accurate reporting of depressive symptoms, although the number and/or depth of questions may vary across different tests.

### 4.1. Progress in the Development of SCC-DBS

Different papers have referred to the SCC and similar regions under different names, e.g., the subcallosal cingulate gyrus (SCG), the subgenual cingulate, as well as Brodmann areas. Different historical names allow for different historical context. The subgenual cortex is used more interchangeably with the term Brodmann Area 25, named after Korbinian Brodmann. The subgenual cortex is located in the cingulate region as a narrow band in the caudal portion of the subcallosal area adjacent to the paraterminal gyrus. By comparison, the SCG is comprised of Brodmann areas 25, 24, and 32 [36]; SCG circuits; and limbic structures. The SCG is pivotal in mood, learning, reward, and memory [37] and has been implicated as an aberrant region in MDD. As the SCC can be effectively targeted by antidepressants, this makes the SCG a potential target of DBS against TRD [38,39]. Table 1 and Table 2 list 39 clinical studies on the treatment efficacy of SCC-DBS for TRD.

The first evidence-based clinical study on SCC-DBS was published by Mayberg et al. in 2005 [19]. Among six patients with an average of 5.6 years of major depressive episode (MDE), four responded to the treatment, but three remitted or nearly remitted during the stimulation, even without changing medications. The authors found that the metabolic activity in the SCC normalized from a hyperactive state and was accompanied by reduced local blood flow as detected by Positron Emission Tomography (PET) [19]. In a study from 2003 to 2006 by Lozano et al. on chronic DBS in 20 patients with an average of 6.9 years of current MDE, 11 patients responded, but seven remitted [72], which was similar to the response and remission rates of Mayberg et al. In a 3.5-year follow-up study, the response rate was consistent across time points, but the remission rate increased from 18.8% to 42.9% at the last visit [18]. Both studies reported changes in structures distal to SCC after DBS, which explains the persistent response throughout the DBS treatments [18,72].

In a case report by Neimat et al., a 55-year-old female TRD patient who relapsed after a subgenual cingulotomy, achieved sustained remission for up to 30 months with SCC-DBS treatment [74]. In a case reported by Guinjoan et al. in 2010, a 60-year-old male TRD patient responded to unilateral SCC-DBS in the right hemisphere, but unilateral stimulation in the left hemisphere worsened his mood. This is in line with the asymmetrical response to antidepressants in the SCC region. However, the authors noted a further study was needed with more patients to validate the effects of unilateral stimulation on mood enhancement [68].

Similarly, in a preliminary study in 2012 by Puigdemont et al. on eight patients with an average of 6.3 years of current MDE, they found that five patients responded at the end of the 12-month DBS, but three out of four final remitters remitted after 3 months of DBS [67]. Their cognitive functions were not exacerbated and their memory functions were actually improved in cognitive assessments in 2015 [56]. Concurrently, a clinical study conducted in three different medical centers also reported similar efficacies of SCC-DBS, suggesting that DBS has reliable stimulation effects. Among 21 patients with an average of 5 years of current MDE, 13 responded to the treatment and the rest performed better than at baseline by the end of the study, although one patient committing suicide by medication overdose [66].

### 4.2. Remission Rates

Some previous studies reported higher initial response and/or remission rates compared to more recent studies [19,67,72]. In the study by Perez-Caballero et al., they suggested that electrode insertion-induced inflammation could affect response and remission rates. Four of the eight recruited patients took non-steroidal anti-inflammatory drugs (NSAIDs), which resulted in a diminished antidepressant response toward DBS, whereas the other four not taking NSAIDs gradually responded and remitted. The authors also analyzed the role of inflammation in the early DBS response in rats [59], which is discussed in a later section of this review. A later study in 2015 by Puigdemont et al. reported that remission was maintained in four out of five remitted patients in the 3-month active stimulation group, whereas only two patients remitted in the sham stimulation group. They concluded that continuous active stimulation was important in maintaining the therapeutic effect [55]. This was supported by an earlier case of a 27-year-old patient on DBS for 2 years whose symptoms worsened due to battery depletion, but improved again upon battery replacement [69].

Table 2 reflects the different response and remission rates, at 6-month intervals, across the duration of the studies in Table 1. This reporting allows for a cursory longitudinal tracking in understanding how response and remission may change with time. Among the reviewed studies on DBS, the response rate ranged from 18% to 87.5% and remission rate ranged from 10% to 92% (excluding all case studies) across the different time points (see Table 2), which were comparable to earlier clinical studies [19,72]. However, large-scale controlled trials are needed to further validate the efficacy of DBS in patients with TRD. Some predictive markers discovered in these studies could facilitate the selection of more responsive patients and increase the safety of DBS. A noteworthy study by Holtzheimer and Mayberg demonstrated some changes in the response and remission rates with DBS [69]. The authors noted that several months after a response and/or remission in their depressive symptoms, worsening of symptoms was temporarily observed at 16 weeks. They attributed the temporary worsening of symptoms to the difficulty of some patients reintegrating into society. In an earlier study by Lozano et al. in 2008, they also observed a similar occurrence at 4 months. These findings highlight the complexity of treating neuropsychiatric diseases, as the recovery periods are not always consistent and can be affected by different factors.

### 4.3. Significant Challenges in the Development of SCC-DBS

A larger study that aimed to recruit 201 patients was conducted by Holtzheimer et al. in 2017 to further validate the therapeutic effects of DBS [15]. A futility analysis conducted after 90 patients had been recruited showed no significant differences between the DBS and sham groups, leading to the early termination of the study. During a 6-month double-blind trial, no significant differences were found in the response of the DBS group compared to the sham group. However, among 77 patients that received subsequent open-label DBS for up to 2 years, 38 responded and 20 remitted. Holtzheimer et al. offered several explanations for the observed result. First, the patients selected for the study had an average current episode duration of around 12 years, whereas most studies recruited patients with an average current depressive episode of about 3–5 years. Holtzheimer also posited the possibility of suboptimal contact during the first 12 months, further affecting the results. This landmark paper was initially thought to be the death knell for DBS as a treatment for TRD. However, a summit of key academics within the field determined that DBS protocols required further modification and patient recruitment needed refining to better assess the therapeutic effects of DBS for TRD [75]. Considering that multiple other studies showed the efficacy and effectiveness of DBS for TRD, the conference considered several possibilities for the discrepancies in the findings, some conclusions were that DBS was initiated too early before optimal targeting was secured, a lack of specificity and standardization in the improvement of symptoms, high placebo effects typically seen in the treatment of psychiatric disorders, and study design. The heterogeneity of the symptoms of the disease was also emphasized, which suggested that different circuitry might be involved in different individuals. The key conclusions from the summit included that patient selection should be better and more refined, study designs should be either fast to fail or fast to succeed, registries should be established for better subject tracking, and longitudinal data should be collected. The paper stressed that the complexities of the disease were real and better experimental designs were needed to truly reflect the effects of DBS as a treatment for TRD for a better response and remission rate and to allow the elucidation of the mechanistic role of DBS.

### 4.4. Adverse Effects

The safety of SCC-DBS was subsequently assessed following the initial results of the efficacy of DBS in treating TRD. In 2008, McNeely et al. conducted a trial on six patients with an average of 5.6 years of current MDE. They found that mood was significantly improved during the 1-year DBS treatment without serious cognitive deterioration [73]. Moreines et al. found that DBS treatment in both unipolar and bipolar TRD patients with at least 2 years of current MDE improved executive functions and stabilized their memory [76]. Similarly, SCC-DBS for 6 months followed by depression treatment in patients with MDD, who had increased negative emotional processing and/or reduced positive emotions, resulted in reduced negative emotional bias [53]. Martín-Blanco et al. reported that a 52-year-old female had a seizure after 5 weeks of DBS. As severe MDD may predispose patients to seizures, the authors recommended that patients should be evaluated for seizures before administering DBS and parameters might need to be adjusted to within safe ranges [52]. In a study in 2017 by McInerney et al. on 20 unipolar TRD patients with an average of 6.9 years of MDE, they reported that 11 patients responded at the end of the 12-month DBS without further deterioration of cognitive functions. They also found a correlation between verbal fluency and mood improvement, which could be predictive of the DBS response [14]. The side effects reported in this review range from mild symptoms, such as headaches, dizziness and gastrointestinal irritation [43,66], to more severe effects including suicidal ideation and device malfunction [15,67]. This reporting should not discourage the development of therapies. Indeed, many treatments including serotonin-selective reuptake inhibitors have severe side effects, including increased fractures and suicidal ideations [77,78]. In the study of therapies, it is important to report these side effects and to note that these therapies are administered by a professional, whose role is to detect and modulate the therapies accordingly.

### 4.5. Stimulation Parameters

Several studies have attempted to optimize the parameters of DBS for treating mood disorders. As previously mentioned, Eitan et al. reported that high-frequency stimulation (HFS) was more effective at lowering MADRS scores compared with low frequency stimulation [42]. Indeed, the most commonly used stimulation frequency was in the high frequency range of 130–135 Hz, although some studies have tested frequencies between 5 and 185 Hz [19,61] (see Table 1). The pulse width used in DBS also varied greatly across studies. In a study by Ramasubbu et al., they found that a long pulse width of 180–270 μs was effective [61]. However, this study also reported that DBS with a long pulse width caused patients to experience stimulation-induced insomnia, anxiety, confusion, and drowsiness. Previous studies by Lozano et al. and Holtzheimer et al. demonstrated that shorter pulse widths of 30–60 μs led to clinical improvements in depression symptoms without these side effects [66,72]. Indeed, Ramasubbu et al. suggested that longer pulse widths with lower amplitudes and shorter pulse widths with higher amplitudes could produce comparable therapeutic benefits. The amplitude of the stimulating current used in DBS to elicit a therapeutic response also tended to vary across studies. The amplitude is the first parameter to be adjusted when patients do not respond to the treatment. Among 38 clinical studies, the overall current range was 2–8 mA and voltage range was 2.5–10.5 V. The variability in the amplitude underscores the personalized nature of DBS, which requires specific adjustments to achieve individual therapeutic effects.

### 4.6. Electrode Implantation

Several clinical DBS studies have also tried to improve the accuracy of electrode implantation in order to precisely target regions of interest. The pioneering work by Mayberg et al. used PET scans of pathological glucose metabolism to guide the electrode implantation. Riva-Posse et al. used individualized tractography maps based on a group connectome blueprint of past responders to DBS to identify optimal target regions for electrode implantation [49]. Riva-Posse et al. used a four-bundle white matter blueprint, which resulted in good clinical outcomes in eight out of 11 patients, which suggests that the use of this method could improve the precision of implantation. Similarly, Tsolaki et al. investigated the use of FMRIB Software Library (FSL) probabilistic tractography in SCC-DBS [50]. Several studies have used other methods to try to specify the optimal stimulation points. Choi et al. investigated the best contact positions that elicited the best response during intraoperative testing [57]. They used diffusion-directed magnetic resonance imaging and patient-specific tractography maps to guide the implantation. They also used fiber tract probabilistic tractography to determine the putative fiber tract activation in patients, which was used to guide the electrode implantation for the best response, rather than the salient response. Contacts in the left hemisphere were found to produce the best consistent intraoperative response to DBS in seven out of nine patients at 6 months. Smart et al. validated this result in their study using local field potentials following unilateral HFS-DBS [46]. They found that left-sided stimulation evoked broadband effects, compared with right sided stimulation, which evoked only beta and gamma bands. Additionally, a decrease in theta bands was consistently accompanied by behavioral improvements. They concluded that the precision of electrodes in the left-hemisphere was more important and instructive than in the right hemisphere. In contrast, Guinjoan et al. and Howell et al. found that right hemisphere targets were critical for behavioral improvements [44,68]. Guinjoan et al. showed that right unilateral DBS could reverse and remit a patient’s worsening mood induced by left unilateral DBS. Howell et al. showed that right cingulate bundle activation beyond a threshold could protract the recovery. Further research is needed to elucidate the differences in these studies.

### 4.7. Other Responses to DBS

With regard to other potential responses to DBS, recent studies have attempted to characterize non-behavioral evoked responses. Conen et al. identified higher rCBF in patients at baseline and during DBS therapy compared to non-remitters and non-responders [48]. Riva-Posse et al. observed autonomic changes in responders undergoing DBS [41]. In a study by Smart et al., assessing the efficacy of DBS, they found consistent changes in left theta local field potentials, which could provide another consistent parameter to monitor. Based on this finding, they proceeded to adjust the contacts for one non-responder, who was able to achieve a response by the end of the study. In a study by Sankar et al. on responders and non-responders who had previously undergone SCG-DBS, they found that both groups had significant volume differences in the left and average SCG; in the right and average amygdala; and in the left, right, and average thalamus (Sankar et al. 2020). Additionally, non-responders had significantly greater grey matter volume compared to responders and a greater grey to white matter ratio. This important information provides yet more criteria for assessing if a patient might respond to DBS. Expanding the breadth of data obtained during clinical trials has the potential to advise clinicians on the efficacy of DBS, and to help predict non-responders and adjust the stimulation parameters. This will improve patient welfare and allows for a more accurate examination of the mechanisms of DBS in improving depressive-like symptoms.

Furthermore, it would be prudent to use preclinical results to advise clinical cases. In a previous preclinical study, Hamani et al. reported that DBS supplemented with tranylcypromine increased the antidepressant-like response in animals by 20%–30% compared to either treatment alone. They later reported on a patient who relapsed after 4 years of remission following DBS treatment [64]. Based on their previous work, they administered tranylcypromine before the DBS treatment, which allowed the patient to enter remission again.

## 5. Preclinical Studies of Electrical Stimulation in the Medial Prefrontal Cortex in Rodents

Following the success of a number of preliminary clinical studies, several preclinical studies were conducted to investigate the antidepressant-like effects of DBS [79]. The mPFC in rats is widely regarded to be homologous to the SCC in humans. The mPFC together with the amygdala, hippocampus, and hypothalamus controls the stress response, autonomic functions, and cognition in rats [80,81,82,83]. Using PET imaging, glucose metabolism was observed to normalize in the mPFC from a hyperactive state following DBS, which was similarly observed in the SCC after 1 h of DBS [84]. However, the homology between subdivisions of mPFC and SCC is still under debate. The vmPFC can be further subdivided into the infralimbic (IL) and prelimbic (PrL) regions. Although there are overlaps, the IL and PrL innervate different regions to different extents, including the lateral hypothalamus, dorsal raphe nucleus, and amygdala, among efferent regions [85,86,87]. The PrL has been shown to innervate to important limbic regions associated with SCC projections [87]. Meanwhile, the infralimbic cortex (IL) is believed to be structurally homologous based on comparisons involving tractography analysis [88,89,90]. Some assert that the whole ventromedial prefrontal cortex (vmPFC) is homologous to the SCC [91,92]. Others assert that the vmPFC is functionally distinct from the dorsal medial prefrontal cortex [70,91]. Nevertheless, the vmPFC is generally regarded as homologous to BA25, although a thorough understanding of specific correlations remains to be seen. Table 3 lists 29 preclinical studies on the effects of vmPFC-DBS on animal behaviors.

### 5.1. vmPFC Stimulation

Hamani et al. published the first preclinical study of vmPFC-DBS in rats in 2010. They used the forced swim test (FST), which models “helplessness” in animals including anxiolytic-like and anti-anhedonic-like behavior. They found DBS reduced the immobility score in FST, indicating antidepressant-like effects. The authors attributed the behavioral changes to serotonergic function in the dorsal raphe nucleus (DRN) as lesions in this structure abolished the behavioral effects in FST [91]. Another animal study in 2012 found the optimal stimulation frequency and amplitude of vmPFC-DBS was 130 Hz and 200 µA that produced anti-anhedonic-like effects and produced a charge density similar to DBS in humans [93]. They found a lesion in the DRN abolished the higher sucrose consumption due to DBS, even with a normal hippocampal brain-derived neurotrophic factor (BDNF) profile. They postulated that an interaction between BDNF and neurochemical substances potentiated the antidepressant-like response [92]. The anti-anhedonic-like effects of DBS were also supported in studies by Rea et al. and Edemann-Callesen et al. They conducted an intracranial, self-stimulation paradigm in Flinders sensitive line and Flinders resistant line rats to assess reward-seeking behaviors, which demonstrated that the anti-anhedonic-like effect of vmPFC-DBS was independent of the dopaminergic reward system [94,95]. Bruchim-Samuel et al. reported that modulation of the ventral tegmental area could prolong the behavioral changes. They found that intermittent acute patterned stimulation administered to the ventral tegmental area of Flinders sensitive line rats resulted in antidepressant-like and anti-anhedonic-like behaviors [96]. Strikingly, a study by Bregman et al. in 2018 found that the antidepressant-like effect of DBS was serotonin transporter-independent. This could be of benefit to some patients with a mutated serotonin transporter-promoter gene (5-HTTLPR), which underlies the poor response to conventional selective serotonin re-uptake inhibitors that target serotonin transporters [97].

Beside changes in neurochemical and neurotrophin profiles, neuroplasticity changes induced by DBS have also been investigated. For instance, Bambico et al. reported increased hippocampal neurogenesis and BDNF levels after vmPFC-DBS, which led to anti-anhedonic-like behaviors, but was not sufficient for an overall antidepressant-like effect [98]. Correspondingly, Liu et al. found a correlation between vmPFC-HFS and hippocampal neurogenesis and improvements in short- and long-term memory in middle-aged rats. This suggests that DBS has therapeutic potential in age-dependent memory deficits [99].

### 5.2. Other Brain Targets

As preclinical studies have progressed, several brain targets of DBS have been established. Hamani et al. in 2014 demonstrated that DBS in the nucleus accumbens induced a similar antidepressant-like effect to DBS in the vmPFC, even though the stimulations modulated different circuits. This may contribute to more customized stimulation targeting based on the patient’s symptoms [100]. Bregman et al. reported that the HFS of the medial forebrain bundle induced antidepressant-like behaviors in the FST [101]. Interestingly, this antidepressant effect was not mediated by increases in either serotonin or dopamine release in the nucleus accumbens. Lim et al. in 2015 emphasized that only HFS of the vmPFC led to anti-anhedonic-like effects and these pronounced antidepressant-like effects were induced by modulating the activity of serotonergic neurons in the DRN [102]. However, the authors did not investigate the effects of different stimulation parameters on depressive-like behaviors in various DBS targets. The study by Etiévant et al. supported the modulation of DRN by DBS and added that glial integrity was a prerequisite to the antidepressant-like outcome [103]. In another study, mice subjected to chronic social defeat stress were administered 7 days of 5-h vmPFC-DBS, which resulted in increased social interactive behavior accompanied by DRN modulation [104]. Interestingly, a recent study demonstrated that the potentiation of the anxiolytic response to vmPFC stimulation was associated with exposure to an enriched environment. This indicates that an enriched living environment can facilitate the beneficial effects of DBS intervention [105]. Creed et al. conducted DBS on the entopeduncular and the subthalamic nuclei to compare antidepressant-like effects [106]. Chronic Subthalamic nucleus DBS was reported to impair performance in the learned helplessness task, with no significant effects in anxiety tests. These results were associated with decreased hippocampal BDNF and TrkB mRNA. Interestingly, entopeduncular nucleus DBS did not increase depressive-like behavior in the learned helplessness task, indicating a superior target over the subthalamic nucleus for the treatment of depressive-like behaviors. Meng et al. reported reductions in depressive-like behaviors in animals stimulated in the lateral habenula; this observation was associated with elevations in dopamine, norepinephrine, and serotonin in both blood serum and in the hippocampus [107].

### 5.3. vmPFC-Linked Modulation of Other Structures

Other structures have been found to be modulated by vmPFC-DBS. For example, Lim et al. reported that activation of the medial subthalamic nucleus contributed to antidepressant-like behavior [108]. In a rat model of post-traumatic stress disorder, IL-DBS reduced firing in the basolateral amygdala, which attenuated fear and produced a slight anxiolytic-like effect [109]. A recent study showed that DBS resulted in elevated spontaneous firing of noradrenergic locus coeruleus neurons and strengthened the coherence between the prefrontal cortex and locus coeruleus. The latter was protective against stress and was responsible for the antidepressant-like effect seen in FST [110]. On the other hand, Insel et al. reported that there was reduced communication between IL and ventral hippocampus in rats after 10 days of 8-h IL-DBS and such coherence was higher in depressed subjects [111]. Jiménez-Sánchez et al. in 2016 reported two studies on acute IL-DBS in naive and olfactory bulbectomized rat models. In naive animals, IL-DBS induced antidepressant-like behaviors and increased prefrontal glutamate efflux, which activated the α-amino-3-hydroxy-5-methyl-4-isoxazolepropionic acid receptor (AMPAR) to modulate DRN output [81]. In olfactory bulbectomized rats, similar changes were noted in the prefrontal serotonergic and glutamatergic output with the activation of AMPAR and antidepressant-like behaviors [33].

### 5.4. Synergism with Other Treatments

Antidepressant-like effects in different DBS paradigms are leading to some advancements in the field. One such investigation by Laver et al. in 2014 examined the use of augmentation agents such as buspirone, risperidone, and pindolol to enhance DBS efficacy. However, these agents did not increase the antidepressant response of the rats receiving DBS treatment, when compared to those co-administered monoamine oxidase inhibitors in previous studies [64,112]. It is possible that a response may become evident in clinical trials. Perez-Caballero et al. in 2014 reported an interesting early response to stimulation, in which sham-treated rats had reduced immobility and increased swimming in FST at weeks 1 and 2, but not at week 6 post treatment. They reasoned that this was caused by insertion-induced inflammation as pretreatment by indomethacin reduced the expression of pro-inflammatory mediators (TNF-α, COX1, COX2) and reversed the antidepressant-like behaviors in sham-treated animals [59]. Rummel et al. in 2016 reported that chronic continuous HFS did not have more benefits than chronic intermittent stimulation in treatment-resistant rats with congenitally learned helplessness [113].

### 5.5. Other Biological Parameters Modulated

Similar to the research direction of clinical studies, preclinical studies have also attempted to characterize other biological parameters of DBS, including more precise electrode implantation. Lehto et al. characterized real-time fMRI responses in the brain following DBS, and found strong connectivity between the vmPFC and amygdala, which validated vmPFC as a target region [114]. Perez-Cabalerro et al. used PET scans to study the immediate effects of electrode implantation. They found that metabolism was decreased locally (vmPFC), but was increased in ventral regions, including dorsal and ventral hippocampus, piriform and insular cortex, nucleus accumbens, ventral tegmental area, ventral pallidum, hypothalamus, and the preoptic area [115]. This was in agreement with other studies on the effect of DBS on depressive-like behavior, but it is noteworthy to see these effects simply via electrode insertion.

Preclinical studies have progressed from studying the behavioral effects of DBS to understanding the accompanying cellular and molecular changes, be they local or distal nodes in the neurocircuitry. However, issues concerning the rodent homologs of SCC and the effect of stimulation in the subdivisions of vmPFC have yet to be resolved and are discussed in the later part of the review.

## 6. Potential Mechanisms of Stimulation-Induced Antidepressant-Like Activities

Several preclinical studies reported that DBS modulates neuronal activities in different brain regions, leading to antidepressant-like behaviors (Figure 1A). The network-wide cellular and molecular changes caused by vmPFC-DBS can be classified into neuroplasticity-dependent and -independent changes (Figure 1B). Neuroplasticity-dependent effects included neurogenesis, increased synaptic plasticity, enhanced neurotrophin signaling, and potential activation of glial cell-mediated changes, whereas neuroplasticity-independent effects included changes in serotonergic (5-HT) and glutamatergic neurotransmission patterns, either locally or in distal structures. Other changes outside the scope of this review might also be relevant.

### ***Neuroplasticity-Dependent Effects of Electrical Stimulation*** 

#### *(i) Neurogenesis is a Long-Term Cellular Change Brought About by Electrical Stimulation* 

Post-mortem studies, pharmacological analyses, and electroconvulsive therapy reports have led to the neurogenic hypothesis of the pathogenesis of depression, whereby atrophy and apoptosis of hippocampal neurons correlated with depression and neurogenesis induce antidepressant-like effects [120,121]. As CA1 and subiculum in the hippocampus project substantially to the IL and the latter feeds back to the hippocampus via the relay nucleus reuniens in the thalamus [102,122], vmPFC-DBS induces a corollary of hippocampal neuromodulation that may mediate the antidepressant-like outcome. Etiévant et al. found that there was increased neurogenesis in the dentate gyrus of the dorsal and ventral hippocampus in rodents after 1-h IL-DBS, as detected by positive BrdU cells, and this was accompanied by reduced immobility in FST [103]. Similarly, Liu et al. reported proliferation of neuronal progenitors after chronic vmPFC-DBS, as demonstrated by increased positive BrdU and Dcx cell counts, as well as upregulated expressions of genes related to neurogenesis (NeuN, Syn, Dcx, Nes) and neuronal differentiation and protective functions (Angpt2, S100a4). These results were correlated with enhanced memory function, which may serve as another indication of vmPFC-DBS [99]. Bambico et al. confirmed that new cells with mature neuronal phenotype were found in the hippocampus after vmPFC-DBS, as detected by BrdU and NeuN co-expression. They also reported that temozolomide-induced reduction of these cells led to a longer latency to feed in a novelty-suppressed feeding test, but did not significantly change immobility in FST. This prompted the authors to further examine the anxiolytic-like and anti-anhedonic-like effects of vmPFC-DBS. In contrast, Winter et al. showed that 1 h of vmPFC-DBS with established DBS parameters in rodents did not increase the percentage of BrdU and Dcx double-stained cells in the dentate gyrus compared to the control [123]. Although research on the interaction between neurogenesis and substrates such as serotonin is ongoing, BDNF may be required to exert this antidepressant-like effect [98]. Neurogenesis is a widely investigated mechanism of DBS and these results indicate positive effects in the hippocampal region.

#### *(ii) Synaptic Plasticity is Altered More Rapidly by Electrical Stimulation than by Neurogenesis* 

Disruption in synaptic functions and signaling are implicated in the pathophysiology of MDD, considering their importance in neurotransmission and ultimately, in cell survival [20]. Chronic stress, a risk factor of MDD, was shown to cause retraction of dendrites in the medial prefrontal cortex [124] and CA3 of the hippocampus [125] in rodents. In the latter, the long-term potentiation (LTP) of synapses was compromised, affecting long-term memory formation [126]. Regarding the changes in synaptic plasticity caused by vmPFC-DBS, Liu et al. reported denser secondary dendritic spines in the dentate gyrus, as demonstrated by upregulated Syn expression correlated with Nes and Dcx. The authors also reported a slight (1.2 fold) increase in hippocampal BDNF gene expression, a regulator of synaptic plasticity [99]. More recently, Bezchlibnyk et al. found that 1 h of IL-DBS resulted in longer dendritic length and branch points localized in the basal and apical regions of hippocampal CA1 neurons, respectively. These results indicated that the acute stimulation stressed the indispensable connections between the hippocampus and vmPFC, which may have implications in MDD and its treatment [127].

Chakravarty et al. found that 5 days of 6-h vmPFC-DBS daily in 9-week-old C57Bl/6 mice resulted in a larger hippocampal volume and increased hippocampal synaptic density, as indicated by upregulated synaptophysin expression, a presynaptic marker [128]. Similarly, Veerakumar et al. found that chronic vmPFC-DBS in transgenic mice increased dendrite length and complexity of the 5-HT DRN neurons and upregulated the expression of postsynaptic markers synaptophysin and PSD-95 [104]. Moreover, Etiévant et al. reported synaptogenesis in the DRN, as indicated by higher expressions of PSD-95 and synapsin. This may explain the prolonged DRN neuronal activation during and after vmPFC-DBS, leading to an antidepressant-like effect [103]. According to earlier reviews, dendritic spines can respond swiftly and provide surfaces for synapse formation [126,129]. Given the more dynamic properties of synapses compared to neurogenesis, synaptic plasticity may serve as an early indicator of vmPFC-DBS efficacy. More preclinical studies characterizing the dynamics of synaptic plasticity under vmPFC-DBS are anticipated.

#### *(iii) Neurotrophin Signaling Underlies the Antidepressant-Like Effect of Electrical Stimulation* 

The neurotrophin BDNF is important in synaptic regulation, neuronal survival, and differentiation of new neuron terminals in the adult brain [130,131,132]. Preclinical studies reported that depressive-like rats subjected to chronic unpredictable stress [92,98] or olfactory bulbectomy [33] had lower BDNF levels, whereas DBS increased BDNF levels, thus preventing the development of depressive-like behaviors. Extracellularly, pro-BDNF is cleaved by tissue plasminogen activator/plasmin to form mature BDNF. The high-affinity tropomyosin-related kinase B (TrkB) receptor is activated by BDNF [133], leading to downstream phosphorylation of kinases, including protein kinase B (Akt) and extracellular signal-regulated kinases (ERK), which are important mediators of anti-apoptosis and proliferation, respectively [134]. Moreover, BDNF-TrkB triggers Serine 133 phosphorylation of transcription factor cAMP response element binding (CREB), leading to the formation of the dimer. The phosphorylated CREB dimer forms a larger transcriptional complex and alters multiple gene expressions including BDNF itself [135]. Encouragingly, Jiménez-Sánchez et al. showed that IL-DBS administered to olfactory bulbectomized rats activated these signaling pathways, as demonstrated by lowered Akt/pAkt, ERK/pERK, and CREB/pCREB ratios during 1 h of stimulation that increased again after stopping the stimulation, which was similar to the expression pattern of BDNF [33]. Further molecular studies are needed to characterize the action of vmPFC-DBS toward different targets in this signaling cascade.

#### *(iv) Potential Involvement of Glial Cells in Mediating the Outcome of Electrical Stimulation* 

Glial cells may be involved in the pathogenesis of depression, as revealed by post-mortem studies of MDD patients, which found lower densities in the prefrontal cortex and amygdala, but increased levels in the hippocampal hilus [136,137,138]. The latter may be activated as a result of neuronal injury and decreasing neuronal populations [139,140]. Glial cells metabolically support neurons and regulate glutamate synthesis and thus, regulate synaptic plasticity. They may be modulated by DBS to potentiate the therapeutic effects [136]. This mechanism was supported in a study by Etiévant et al., which found that glial lesion by L-alpha-aminoadipic acid injection diminished antidepressant-like behaviors, hippocampal neurogenesis, and LTP induced by IL-DBS [103]. These findings led to the hypothesis that the neuronal-glial relationship is a determinant of the antidepressant-like efficacy of DBS, but this requires further study. Perez-Caballero et al. also studied the effects of electrode implantation and analgesic supplements [115]. They found that implantation with non-NSAID analgesic treatments, like tramadol and morphine, did not ameliorate the anti-depressant effects of the electrode implantation. This observation was accompanied by an increase in glial marker GFAP-positive cells. This finding suggests that the supplementation of non-NSAIDs postoperatively could improve the comfort of patients.

### ***Neuroplasticity-Independent Effects of Electrical Stimulation*** 

Besides modulating neuroplasticity-dependent mechanisms, DBS may manipulate some neuroplasticity-independent pathways to induce antidepressant-like effects. In a chronic mild stress model, depressive-like behaviors developed without significant deterioration of hippocampal neurogenesis or neuronal survival [141]. There are two likely inter-related neurotransmission systems that potentiate DBS efficacy, namely serotonergic and glutamatergic systems.

#### *(i) An Alternative Action of the Serotonergic System by Electrical Stimulation* 

Results from preclinical studies have established an important role of the vmPFC-DRN axis and downstream 5-HT neurotransmission in the treatment of depression. Hamani et al. first reported that 5-HT neurotransmission was augmented by DBS, as shown by a four-fold increase in hippocampal 5-HT after 1 h of vmPFC-DBS [91]. The authors also suggested a relationship between the integral 5-HT system and DBS efficacy, as 5-HT depletion induced by DRN lesions with 5,7-dihydroxytryptamine injection diminished the antidepressant-like effects of vmPFC-DBS [91,92]. Similarly, a study by Perez-Caballero et al. showed that the administration of para-chlorophenylalanine ester impeded 5-HT biosynthesis and diminished the antidepressant-like behaviors in early DBS among IL sham-treated animals [59]. Interestingly, Volle et al. showed that DBS and fluoxetine could rescue the 5-HT system via different mechanisms [119]. Both treatments increased the amount of 5-HT at the end of the chronic treatments, but chronic fluoxetine treatment was associated with decreased expression of 5-HT_1A_ in the prefrontal cortex and the hippocampus, whereas chronic DBS increased 5-HT_1B_ expression in the prefrontal cortex, globus pallidus, substantia nigra, and raphe nuclei.

A study by Veerakumar et al. in a transgenic mouse model of chronic social defeat stress revealed normalization of 5-HT neuron excitability in DRN after vmPFC-DBS [104]. Moreover, Jiménez-Sánchez et al. found increased prefrontal 5-HT efflux after 1 h of IL-DBS in olfactory bulbectomized rats [33] and in naive rats [81]. Etiévant et al. also found spontaneous DRN 5-HT neuron activity increased with IL-DBS [103]. Strikingly, an electrophysiological study performed by Srejic et al. showed that IL-DBS decreased the firing rate of DRN neurons, including serotonergic subtypes via the activation of GABAergic interneurons and possibly by the inhibition of excitatory glutamatergic neurons that modulate the firing of 5-HT [142]. Hence, the positive response to DBS could be enhanced by more selective targeting of the neuronal population by pharmacological adjuncts or coupling with optogenetic techniques. A study by Bregman et al. in 2018 using a serotonin transporter knockout mouse model found that DBS increased hippocampal 5-HT concentration, despite mice responding poorly to fluoxetine, a conventional selective serotonin reuptake inhibitor that acts on serotonin transporter [97]. These findings revealed a novel antidepressant-like activity of DBS involving the 5-HT system primarily in the DRN [79].

#### *(ii) Glutamatergic Neurotransmission is a Promising Target of Electrical Stimulation* 

Jiménez-Sánchez et al. showed that there was enhanced prefrontal glutamatergic efflux together with changes in the local 5-HT profile [33,81]. The administration of AMPAR agonist and antagonist and subsequent FST showed that the increased glutamate led to antidepressant-like behaviors in animals [81]. The authors also found increased synthesis of the GluA1 subunit of AMPAR and postulated that their postsynaptic membrane insertion may explain the antidepressant-like outcome after 1 h of IL-DBS [33]. The activated glutamate output from the medial prefrontal cortex and frontal cortex enhanced 5-HT neuronal firing in the DRN [143], resulting in the antidepressant-like effect. However, Etiévant et al. argued that their activation was attributed to increased synaptogenesis in the DRN as previously described [103]. Nevertheless, Lim et al. hypothesized that a glutamatergic projection from the vmPFC to the medial subthalamic nucleus [144] may account for the antidepressant-like effects of vmPFC-DBS, as seen by increased c-Fos-immunoreactive cells in the medial subthalamic nucleus, increased sucrose consumption, and reduced immobility duration in FST [108]. With the emergence of glutamate-targeting pharmacotherapy [81], the ability to modulate glutamatergic transmission of DBS would add to the therapeutic novelty.

## 7. Concerns and Limitations of the Electrical Stimulation Studies

The small sample sizes in several clinical studies might compromise the credibility of the DBS efficacy, even in studies with similar recruited DBS subjects or consistent outcomes [18,56]. Most of the clinical studies were open-label, which means the responses could be prone to the placebo effect, despite the early response characterized by Perez-Caballero [59]. Efficacy of DBS treatment could be overestimated, unless countered by long stimulation, randomization, and blinding [18,145], such as double-blinded and sham-controlled studies [55]. A major criterion in preclinical studies is that they should mimic clinical studies. As it is unfeasible to stimulate animals for 24 h a day as in clinical designs [128], the scheduling of the stimulations and behavioral assessments will thus be relevant to the validity of the outcomes. Stimulation during behavioral tests will be most similar to clinical studies, but this may interfere with the physiological functions of the animals [79]. Besides, DBS is normally carried out in animals for relatively short periods and the effects might not correlate well with chronic stimulation [33]. Some preclinical studies were conducted in naive animal models and would not be compatible with clinical trials as TRD patients will be recruited exclusively in clinical settings [33,93]. Moreover, carry-over effects and lesion effects may interfere with the results in both settings. Carry-over effects refer to behavioral or neurochemical changes after DBS ceases. This needs to be counteracted by a washout period to allow the subjects to resume their baseline physiological states before the next stimulation [79]. Lesion effects occur where responses are observed after electrode implantation [19,60,72]. This needs to be differentiated from true responses observed in preclinical studies by sham-treatment [67], otherwise, the therapeutic effect will be over-estimated. Generally speaking, care must be taken in the design of experiments and data analysis of preclinical studies to increase their translational value to clinical studies.

## 8. Prospective Approaches to Enhance Deep Brain Stimulation Safety and Efficacy

Clinical response to SCC-DBS and various predictors can facilitate precise patient selection and customize the stimulation targets, thereby yielding maximal therapeutic outcomes with minimal adverse effects. For example, a lower baseline frontal theta cordance and incremental increase in the early stage of DBS indicates a clinical response [63]. Efforts have been made toward a more standardized approach to localize SCG in DBS responders [70]. Recently, real-time recording of the local field potential at the site of electrode implantation coupled with electroencephalogram have revealed network-wide clinical changes in DBS, which may improve the surgical precision [146]. Tractography-guided localization of electrodes, being more customized and precise, can improve the response rate [49]. A rechargeable DBS system should also be considered for long-term stimulation to reduce the need for surgery to replace batteries [17]. Lastly, given the high cost and invasiveness of DBS, more stringent regulation and evidence from randomized controlled studies are necessary to justify the benefits in TRD patients [147,148].

## 9. Conclusions

Major depressive disorder is a debilitating psychiatric condition, which is affected by treatment resistance. Although safety concerns were raised on the risks of ablative treatment, it paved the way for deep brain stimulation as an adjustable therapy against depression. This review summarized the efficacies of deep brain stimulation in the subcallosal cingulate, one of the most extensively studied targets of stimulation, and in the ventromedial prefrontal cortex, which is the rodent homolog. Research on DBS initially focused on symptomatic relief. As the decades have progressed, studies have started to branch out and utilize modern technology to improve targeting of brain regions and to investigate a broader list of symptoms in patients. This has allowed us to better understand the impact of DBS on underreported parameters, such as heart rate, skin conductance, and brain waveforms. Additionally, preclinical research has expanded our understanding of the molecular factors modulated by stimulation. Besides the local effects, DBS has been shown to modulate distal structures, which can involve numerous projections to and from the stimulated targets, and can contribute to the antidepressant effects. This review also described some of the neuroplasticity-dependent and -independent changes brought about by DBS. Progress in different areas of research has helped lay the groundwork for the next wave of DBS research investigating more targeted and more effective applications of DBS for treating MDD. Last but not least, with further customization, more precise approaches, and more stringent regulation, it is anticipated that deep brain stimulation has great promise to treat severe, refractory depressive disorders in the near future.

## Figures and Tables

**Figure 1 jcm-09-03260-f001:**
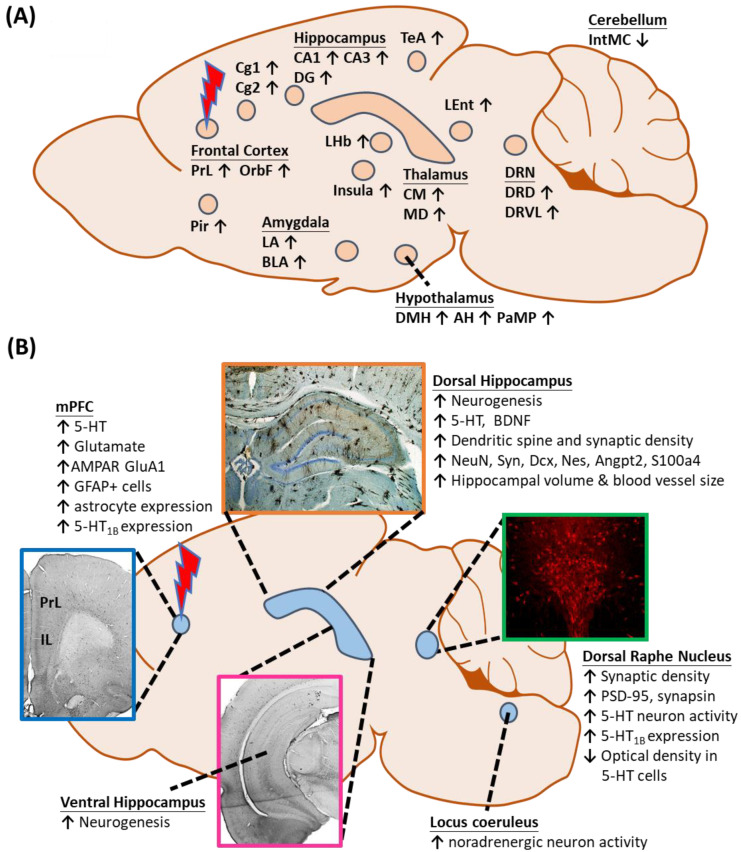
(**A**) Changes in local and distal neuronal activity after electrical stimulation of the ventromedial prefrontal cortex. (**B**) Neuroplasticity-dependent and -independent changes in different structures following vmPFC-DBS. Abbreviations: AH, anterior hypothalamus; AMPAR, α-amino-3-hydroxy-5-methyl-4-isoxazolepropionic acid receptor; BDNF, brain-derived neurotrophic factor; BLA, basolateral amygdaloid nucleus; Cg1,2, cingulate gyrus area 1, 2; CM, centromedial thalamic nucleus; DG, dentate gyrus; DMH, dorsomedial hypothalamus; DRD, dorsal raphe nucleus, dorsal part; DRVL, dorsal raphe nucleus, ventrolateral part; IntMC, interposed cerebellar nucleus, magnocelluar part; LA, lateral amygdaloid nucleus; LEnt, lateral entorhinal cortex; LHb, lateral habenula; MD, mediodorsal thalamic nucleus; mPFC, medial prefrontal cortex; OrbF, orbitofrontal cortex; PaMP, paraventricular hypothalamic nucleus, medial parvicellular; Pir, piriform cortex; PrL, prelimbic cortex; TeA, temporal association area; and 5-HT, serotonin.

**Table 1 jcm-09-03260-t001:** A list of clinical studies on deep brain stimulation of the human subcallosal cingulate.

Authors	Main Inclusion Criteria	No. of Patients	Stimulation Target & DBS Design	Stimulation Parameters	Clinical Evaluation	Major Outcomes	Adverse Effects
Sankar et al. 2020 [40]	TRD MDE; Current MDE ≥ 12 months; HRSD-17 score ≥ 20; Non-responsive (NR) ≥ 4 antidepressant therapies; HRSD-17 score ≥ 20.	27	Target SCG	Implantation Bilateral Stimulation Monopolar Frequency 130 Hz Amplitude 3–6 V Pulse Width 90 μs	Volumetric analysis, Whole brain grey and white matter analysis	Left and average SCG volume significantly higher in responders compared to non-responders. Right and average amygdala volume significantly higher in responders compared to non-responders. Left, right, and average thalamus volume significantly higher in responders compared to non-responders. Brain grey matter volume significantly lower in responders compared to non-responders. Ratio of grey to white matter volume significantly higher in responders compared to non-responders.	N.A.
Riva-Posse et al. (2019) [41]	TRD MDE; Current MDE ≥ 12 months; HRSD-17 score ≥ 20; Non-responsive (NR) ≥ 4 antidepressant therapies; HRSD-17 score ≥ 20; GAF < 50.	9	Target SCC Study Design Intraoperative Sessions: 6 min session of 3 min stimulation ON and 3 min stimulation OFF. Number of sessions: 12 trials (one at each of the eight available contacts. Four per hemisphere, plus four sham trials). Sham-controlled, double-blinded trials (one case w/single blind trial)	Implantation Bilateral Stimulation Monopolar Frequency 130 Hz Amplitude 6 mA Pulse Width 90 μs	ECG, EDA, MRI Volume of Tissue Activated, Structural Connectivity Analysis	Autonomic changes with SCC-DBS correspond to salient behavioral responses. Distant effects of SCC-DBS in the midcingulate cortex. Increase in heart rate was only seen with left SCC-DBS. No significant relationship with skin conductance. These findings aid in the optimal selection of contacts and parameters in SCC-DBS surgery.	N.A.
Eitan et al. (2018) [42]	Both sexes, 21–70 years; Non-psychotic MDD; First MDE onset before 45 years old with current MDE ≥ 12 months; NR ≥ 4 antidepressant therapies; MADRS ≥ 22; GAF < 50; MMSE > 24; No changes in current antidepressant treatments ≥ 4 wks prior to study.	9	Target BA25 Study Design Double-blind, randomized. Two groups: High- OR low-frequency DBS for 12 months from 1 month after electrode implantation.	Implantation Bilateral Stimulation Monopolar Frequency 20 Hz or 130 Hz Amplitude 4–8 mA Pulse Width 91 μs	MADRS, HRSD-17, QIDS-SR, Q-LES-Q, GAF, HAM-A, CGI, PGI, CANTAB battery.	Four out of nine patients responded ^◊^ at the end of DBS (≥40% reduction in MADRS from baseline). The effect of DBS at 6–12 months was higher than DBS at 1–6 months. High-frequency DBS showed higher efficacy than low-frequency DBS. Non-responders crossed over after first 6 months of DBS.	Severe One patient overdosed on medication (dothiepin and valium).
Merkl et al. (2018) [43]	Diagnosed MDD and disease lasted for >2 years; HAMD-24 score ≥ 20; ATHF Score ≥ 3; TRD: NR ≥ 2 antidepressant therapies; Failed to respond to antidepressants and ECT; No changes in current antidepressant treatments ≥6 wks prior to study.	8	Target SCG Study Design Randomized; Two groups: sham-DBS (delayed onset) OR non-delayed onset group for the first 8 weeks in a blinded manner. Open-label DBS afterwards for up to 28 months.	Implantation Bilateral Stimulation Monopolar Frequency 130 Hz Amplitude 5–7 V Pulse Width 90 μs	HAMD-24, BDI, MADRS.	Three out of eight patients responded ** after 6 months DBS; three out of seven with the same criteria after 12 months. Two out of six responded ** at the end of DBS, follow-up at 28 months. Two out of six patients reached remission ^□^ at the end of DBS, follow-up at 28 months. This study showed a delayed response in patients; no significant antidepressant effects between sham and active stimulation compared to baseline.	Non-severe Headache; Pain; Scalp tingling; Dizziness; Light hypomania; Inconvenient movement; Severe NIL
Howell et al. (2018) [44]	Both sexes aged 18–70 years; current MDE ≥ 12 months; NR ≥ 4 antidepressant therapies; HRSD-17 score ≥ 20; GAF score ≤ 50.	6	Target SCC (Cingulum Bundle and Forceps Minor)	Implantation Bilateral Stimulation Monopolar Frequency 130 Hz Amplitude 4 V Pulse Width 90 μs	HDRS-17, MRI Field-cable modeling (non-VTA-based analysis)	All of the subjects responded. Left and right cingulum bundles as well as forceps minor are the most likely therapeutic targets. Right cingulum bundle activation beyond a threshold may protract recovery. Uncinate fasciculus and frontal pole were activated to a lesser extent, may not be necessary for anti-depressive effect of SCC-DBS. Time to a stable response (TSR) was 8–189 days, 1-year HDRS-17 was 2–11. Field cable modeling was more accurate than volume of activated tissue at approximating axonal activation. Overstimulation of CB-DBS can be detrimental to the recovery process.	N.A.
Waters et al. (2018) [45]	Both sexes aged 18–70 years; current MDE ≥ 12 months; NR ≥ 4 antidepressant therapies; HRSD-17 score ≥ 20; GAF score ≤ 50.	4	Target SCC Study Design Single-blinded, Session 3 min	Implantation Bilateral Stimulation Monopolar Frequency 130 Hz Amplitude 3–5 V Pulse Width 90 μs	HDRS-17, EEG	Symptom severity scores decreased. three out of four patients in remission (HDRS-17 ≤ 7). Test-retest reliability across four repeated measures over 14 months met or exceeded standards for valid test construction in three out of four patients for cortical-evoked responses studied.	N.A.
Smart et al. (2018) [46]	TRD patients enrolled from two separate clinical trials for Deep Brain Stimulation. Trial 1 Both sexes aged 18–70 years; Diagnosis of a Major Depressive Episode or Bipolar Type II—current episode depressed, Current episode duration of at least 1 year, Non-responsive (NR) ≥ 4 antidepressant therapies. Trial 2 Both sexes aged 25–70 years; Current depressive episode of at least 2 years duration OR a history of more than four lifetime depressive episodes, Non-responsive (NR) ≥ 4 antidepressant therapies.	14	Target SCC Study Design Double-blinded. Intraoperative behavioral testing: Frequency: 130 Hz Pulse width: 90 μs Current: 6 mA Eight patients continued SCC local field potential.	Implantation Bilateral Stimulation Monopolar Frequency 130 Hz Amplitude 6–8 mA for St. Jude Medical devices, 3.5–5 V for Medtronic devices Pulse Width 90 μs	HDRS-17, MRI, LFP.	11 of 14 patients met the criteria for DBS antidepressant response by 6 months. Of the three 6 month non-responders, one responded after the 6 month study endpoint but without a contact change (Patient 2), one responded after a contact switch in the left hemisphere (Patient 7), and one remained a non-responder (Patient 6). Mean baseline HDRS-17 of 23.8 and SD of 2.8; HDRS-17 of 9.6 and SD of 4.5 at month 6; 19.9 weeks for stable response with SD of 20 weeks. Precision on the left may be more important than precision on the right, which is supported by theta decreases.	N.A.
Choi et al. (2018) [47]	Both sexes aged 18–70 years; current MDE ≥ 12 months; NR ≥ 4 antidepressant therapies; HRSD-17 score ≥ 20; GAF score ≤ 50.	15	Target SCC Study Design Patients went through SCC-DBS, followed by MRI scans.	Implantation Bilateral Stimulation Monopolar Frequency 130 Hz Amplitude 6 mA Pulse Width 90 μs	HDRS-17, MRI, DWI, Volume of Tissue Activated	Significant differences in the pathway activation changes over time between remitters and non-remitters. Non-remitters had significantly larger net changes in their pathway activation connection in both the near and long term relative to the initial plan.	N.A.
Conen et al. (2018) [48]	TRD (unipolar); NR ≥ 4 antidepressant therapies; MADRS Score ≥ 22.	7	Targets SCC followed by Ventral Anterior Capsule, nucleus accumbens (separately, unless patient in remission, and later combined, for non-responding patients). Study Design DBS was applied sequentially for 3 months per region, for a total period ranging from 16–45 months.	N.A.	MADRS, HAM-D 17, GAF.	Remitters had higher regional cerebral blood flow in the baseline prefrontal cortex and subsequent tests when compared to non-remitters and non-responders. Chronic DBS increased prefrontal cortex regional cerebral blood flow. Remitted patients had higher prefrontal cerebral blood flow at baseline.	N.A.
Holtzheimer et al. (2017) [15]	Both sexes aged 21–70 years; Unipolar, non-psychotic MDD First MDE onset before 45 years old with current MDE ≥ 12 months; NR ≥ 4 antidepressant therapies, MADRS Score > 22; GAF < 50; MMSE > 24; No changes in current antidepressant treatments ≥ 4 wks prior to study.	60 (DBS) 30 (Sham)	Target SCC Study Design DBS or sham stimulation 2 weeks after implantation for 6 months in randomized and double-blind manner. Two groups: DBS or sham then both groups received open-label stimulation for 6 months or 2 years.	Implantation Bilateral Stimulation Monopolar Frequency 130 Hz Amplitude 4–8 mA Pulse Width 91 μs	MADRS, GAF HRSD-17, 30-item Inventory of Depressive Symptomatology, QIDS-SR, WSAS, PGI, CGI, QOL, HAM-A.	Insignificant difference in response * between sham and DBS at the end of the 6-month double-blind phase. 38 patients responded * and 20 remitted ^□^ after 6 month DBS. In 2 years of open-label active DBS, 48% achieved antidepressant response and 25% achieved remission—clinically meaningful long-term outcomes.	Severe Eight of 40 events reported related to device or surgery: six infections (in five patients), one skin erosion over the extension wires, and one postoperative seizure.
McInerney et al. (2017) [14]	Current MDE ≥ 12 months; HRSD-17 Score ≥ 20; NR ≥ 4 antidepressant therapies.	20	Target SCG Study Design DBS for 12 months open-label	Implantation Bilateral Stimulation Monopolar Frequency 130 Hz Amplitude 3.5–5 V Pulse Width 90 μs	Wisconsin Card Sorting Task (WCST), Hopkins Verbal Learning Test, Controlled Oral Word Association Test (COWA), Finger Tap Test, Stroop Test, HRSD-17.	Significant reduction in HRSD-17 from baseline to experimental follow-up. Baseline scores differed significantly between responders and non-responders. 11 patients responded ** and nine were non-responders. WCST Test results indicated that the total errors were predictive of responsiveness to DBS. No significant deterioration in cognition and psychomotor speed. Improvements in verbal memory and verbal fluency.	N.A.
Riva-Posse et al. (2018) [49]	Both sexes aged 18–70 years; current MDE ≥ 12 months; NR ≥ 4 antidepressant therapies; HRSD-17 score ≥ 20; GAF score ≤ 50.	11	Target SCC Study Design DBS from 4 weeks after surgery and lasted for 6 months, open-label. Stimulation contacts were changed in non-responders and they were stimulated for 6 more months.	Implantation Bilateral Stimulation Monopolar Frequency 130 Hz Amplitude 6–8 mA Pulse Width 91 μs	HRSD-17	Eight out of 11 responded ** and six remitted ^□□^ after 6 month DBS. Nine responded ** and six remitted ^□□^ after 12 month DBS. Two did not respond throughout the study. Tractography-based surgery reduced variability in the effects of stimulation on patient-specific brain circuitry.	N.A.
Tsolaki et al. (2017) [50]	TRD	2	Target SCC	Implantation Bilateral Stimulation Monopolar Frequency 130 Hz Amplitude 8 mA Pulse Width 91 μs	MRI, DTI, CT, FSL Probabilistic tractography, Volume of Tissue Activated, MADRS.	One patient was a responder (81% change in MADRS score). Responder’s contacts were closer to the Tractography-guided optimized target (TOT), unlike the non-responder.	N.A.
Accolla et al. (2016) [51]	MDD; NR to treatments; Currently in a depressive episode as in DSM-IV Axis I disorders; HAMD-24 score of ≥ 20.	5	Target BA25 Study Design Double-blind. Each homologous electrode pair was activated separately on 5 consecutive days, then antidepressant effects was assessed 24 h later. Open-label DBS for up to 24 months. Pre- and post-DBS MRI images were taken.	Implantation N.A. Stimulation Monopolar Frequency 130 Hz Amplitude 5 V Pulse Width 90 μs	HAMD-24, BDI.	Four out of five patients did not show sustained response ** to DBS (also ≥50% reduction in DBI). One patient responded ** to DBS of the bilateral posterior gyrus rectus instead of the intended target (BA25).	N.A.
Richieri et al. (2016) [52]	Diagnosed MDD; Severe cognitive defects and relapsed after ECT.	1	Target BA25 Study Design DBS at Day 5 after electrode implantation.	Implantation Bilateral Stimulation Bipolar Frequency 130 Hz Amplitude 4.2 V Pulse Width 90 μs	QIDS SR-16	Remitted (QIDS SR-16 3/48) at 1 month after DBS and maintained at the end of DBS.	Seizure
Hilimire et al. (2015) [53]	Both sexes aged 18–70 years; Current MDE ≥ 12 months, Non-responsive (NR) ≥ 4 antidepressant therapies, HRSD-17 score ≥ 20, GAF score ≤ 50.	7	Target SCC Study Design DBS for 6 months, open-label. Behavioral testing and electrophysiological recording (i) before electrode implantation, (ii) after 1 month DBS and (iii) after 6 month DBS.	Implantation Bilateral Stimulation Monopolar Frequency 130 Hz Amplitude 4–8 mA Pulse Width 91 μs	HDRS-17, Emotional self-referential task, EEG recording.	Reduced proportion of negative self-descriptive words compared to baseline after 1 month and 6 month DBS. Significant reduction in P1 amplitude compared to baseline (for negative word self-description) after 1 month and 6 month DBS, and P3 amplitudes at 6 month DBS only Reduced depression severity.	N.A.
Martin-Blanco et al. (2015) [54]	Both sexes aged 18–70 years; current MDE ≥ 12 months; NR ≥ 4 antidepressant therapies; HRSD-17 score ≥ 20; GAF score ≤ 50.	7	Target SCG Study Design Chronic DBS for 9 months on average for clinical stabilization. A PET scan was acquired (i) during active stimulation and (ii) after 48 h of inactive stimulation.	Implantation Bilateral Stimulation Monopolar Frequency 135 Hz Amplitude 3.5–5 V Pulse Width 120–210 μs	HAMD-17, PET	Decreased metabolism in BA24, BA6, caudate putamen after 48 h DBS. This study suggests metabolic changes spread after longer periods of no stimulation. No clinical changes were detected according to HAMD-17.	N.A.
Puigdemont et al. (2015) [55]	Severe TRD; Both sexes aged 18–70 years; current MDE ≥ 12 months; NR ≥ 4 antidepressant therapies; HRSD-17 score ≥ 20; GAF score ≤ 50.	5	Target SCG Study Design Randomized, Double-blind. After stable clinical remission to DBS, patients were allocated to two groups, one with (i) 3 month DBS-ON, then (ii) 3 month sham stimulation (ON-OFF arm) or OFF-ON arm and the other, vice-versa.	Implantation Bilateral Stimulation Monopolar Frequency 130–135 Hz Amplitude 3.5–5 V Pulse Width 120–240 μs	Volume of Tissue Activated, HRSD-17	Active stimulation: four of five patients were remitted patients. Sham stimulation: Only two patients remained in remission, another two relapsed, and one showed a progressive worsening without reaching relapse criteria.	N.A.
Serra-Blasco et al. (2015) [56]	Treatment-Resistant Depression (TRD) Group Resistant to pharmacological treatment; min. stage IV of Thase-Rush scale; HDRS score ≥ 18. First Episode MDD (FE MDD) Group HDRS score ≥ 14; Newly diagnosed MDD.	16	Target SCG Study Design DBS began at 48 h postoperative and ended when each patient had stabilized response for at least three consecutive visits, tests conducted before surgery, and 12 months after DBS treatment.	Implantation Bilateral Stimulation Monopolar Frequency 135 Hz Amplitude 3.5–5 V Pulse Width 120–210 μs	Rey Auditory Verbal Learning Test, Trail Making Tests-A and -B, Wechsler Adult Intelligence Scale III, Tower of London Test, HDRS-17.	FE MDD and TRD saw significant improvements over time in memory. No significant difference was observed in both groups on executive functioning, language, and processing speed. DBS was well tolerated and had no adverse effect on neuropsychological and cognitive function.	N.A.
Choi et al. (2015) [57]	Both sexes aged 18–70 years; current MDE ≥ 12 months; NR ≥ 4 antidepressant therapies; HDRS-17 score ≥ 20; GAF score ≤ 50.	9	Target SCC Intraoperative Sessions: 6 min session of 3 min stimulation ON, and 3 min stimulation OFF. Number of sessions: 12 trials (one at each of the eight available contacts; four per hemisphere, plus four sham trials). Study Design Sham-controlled, Double-blind trials (one case w/single blind trial).	Acute Implantation Bilateral Stimulation Monopolar Frequency 130 Hz Amplitude 6 mA Pulse Width 90 μs	MRI with FSL analysis, Volume of Tissue Activated.	Behavioral switch was apparent to patients within the first minute of the initiation of stimulation and effects were sustained while stimulation remained on. Three common white matter bundles were affected by stimulation: (i) the uncinate fasciculus, (ii) the forceps minor, and (iii) the left cingulum bundle.Seven of nine patients with left hemispheric contact had a response to treatment at 6 months.	
Sun et al. (2015) [58]	NR ≥ 4 antidepressant therapies, Mean HRSD-17 score of 25 (3).	20	Target SCC Session: 100 min, w/15 min break EEG recording sessions/day Session 1: DBS On Session 2: DBS Random (On/Off) Session 3: DBS (Off)	Implantation Bilateral Stimulation Monopolar Frequency 130 Hz Amplitude 2–7.25 mA OR 2–6 V Pulse Width 90 μs	EEG, HDRS-17.	Suppression of gamma oscillations by DBS during working memory performance and the treatment efficacy of DBS for TRD may be associated with the improved GABAergic neurotransmission, previously shown to be deficient in MDD.The present study also suggests that modifying treatment parameters to achieve suppression of gamma oscillations and increased theta-gamma coupling may lead to optimized DBS efficacy for TRD.	N.A.
Perez-Caballero et al. (2014) [59]	18–70 years old with MDE; Resistant to pharmacological treatment and at most, a partial response to ECT; HAMD-17 Score ≥ 18.	8	Target SCG Study Design All patients received chronic DBS within 48 h after implantation. Four patients took NSAIDs for up to 30 days postoperative, four patients did not.	Implantation Quadrupolar Stimulation 135 Hz Amplitude 3.5–5 V Pulse Width 120–210 µs	HDRS-17	At week 1 after surgery, all patients without NSAID prescription responded ** and two remitted ^□□□^; three patients with NSAID responded **, and two remitted ^□□□^. At week 4 after surgery, three patients without NSAID remitted ^□□□^; no patients with NSAID responded **	N.A.
Merkl et al. (2013) [60]	MDD; NR to treatments; Currently in a depressive episode as in DSM-IV Axis I disorder; HAMD-24 score of ≥ 20; HDRS-24 score ≥ 24.	6	Target SCG Study Design DBS on 11–19 days after electrode implantation, 24 h acute stimulation followed by sham stimulation for each of the three electrode pairs. Up to 6 months of chronic stimulation.	Implantation Bilateral Stimulation Monopolar Frequency 130 Hz Amplitude 2.5–10 V Pulse Width 90 µs	HAMD-24, MADRS BDI, TMT-A, TMT-B, CVLT, TAP, Boston Naming Test, Stroop Test, Word Fluency Test.	Non-significant reduction in HAMD-24, BDI, and MADRS scores for acute DBS and sham stimulation. 0/4 contact pair locations showed significant BDI and MADRS improvements. Contact pair 3/7 for Patient 4 saw a 77% reduction in HAMD-24 score and 62% reduction of MADRS score. Reduced HDRS-24, BDI, and MADRS scores at the end of chronic stimulation. Two out of six remissions ^□^ at the end of chronic stimulation.	Mild Headache; Pain; Scalp tingling; Dizziness; Sore throat; Hardware-related; Severe NIL
Ramasubbu et al. (2013) [61]	Aged between 20–60 years; Diagnosed MDD; HAMD-17 score ≥ 20; NR ≥ 4 antidepressant therapies. (Enrolled patients were among the most treatment resistant).	4	Target SCC Study Design Double-blind DBS optimization. Open-label continuous DBS for 6 months after optimization period. Varied parameters for each patient during optimization.	Implantation Bilateral Stimulation Monopolar Frequency 0/5/20/50/130/185 Hz Amplitude 0–10.5 V Pulse Width 0/90/150/ 270/450 μs	HAMD-17, MADRS, HAM-A, CGI.	Postoperative optimization: All four patients showed maximal response at longer pulse widths; three patients experienced a 50% reduction in HAMD-17 score. Longer pulse widths were correlated to short-term improvement. Longer pulse width also induced insomnia, confusion, and drowsiness; improved by turning off stimulation. Chronic stimulation: two patients responded ** at the end of open-label DBS, with longer pulse width. Electrode targets suggested to be individualized, as opposed to standard as in movement disorders, owing to the complexity of cortical gyral anatomy	Mild Anxiety; Drowsiness; Confusion; Insomnia.
Torres et al. (2013) [62]	Type I bipolar depression; Poor response to ECT and pharmacotherapy.	1	Target SCC Study Design DBS from 15 days after implantation and follow-up for 9 months.	Implantation N.A. Stimulation Monopolar Frequency 130 Hz Amplitude 6 mA Pulse Width 91 µs	HDRS-17, BDI, MADRS, GAF, Young Mania Scale.	Scores improved across tests. Psychotic symptoms disappeared. Manic episodes reduced.	N.A.
Broadway et al. (2012) [63]	Both sexes aged 18–70 years; current MDE ≥ 12 months; NR ≥ 4 antidepressant therapies; HRSD-17 score ≥ 20; GAF score ≤ 50.	12	Target SCC Study Design DBS for up to 24 weeks.	Implantation N.A. Stimulation Monopolar Frequency 130 Hz Amplitude 6–8 mA Pulse Width 90 µs	HRSD-17, Frontal and posterior Theta cordance.	Reduced HDRS-17 scores between baseline and the end of DBS among all patients. Six patients had significantly reduced HRSD-17 scores ** at the end of DBS. Increased frontal theta cordance between baseline and week 4 in responders correlated with their decreased depressive state at later time points.	N.A.
Hamani et al. (2012) [64]	TRD; NR to respond to pharmacotherapy, psychotherapy, transcranial magnetic stimulation, ECT, vagus nerve stimulation. Relapsed after receiving 6 month SCC-DBS.	1	Target SCC Study Design Administered tranylcypromine before surgery. DBS for 6 months.	Implantation Bilateral Stimulation Monopolar Frequency 130 Hz Amplitude 2.5 V Pulse Width 90 µs	HAMD-17	Before relapse: SCC-DBS reduced HAMD-17 score from 22 to 9 after 4 month DBS. After relapse: MAOI supplementation restored the therapeutic effect of DBS; HAMD-17 score lowered from 22 to 16 (after 2 weeks), to 8 (after 2 months) and to 9 (after 4 months).	N.A.
Holtzheimer et al. (2012) [65]	Both sexes aged 18–70 years; current MDE ≥ 12 months; NR ≥ 4 antidepressant therapies; HRSD-17 score ≥ 20; GAF score ≤ 50.	17	Target SCC Study Design Intraoperative testing of electrode location for 12 or 17 patients. Stimulation: (i) 4 weeks of sham stimulation, followed by (ii) 24 weeks of open label DBS for 24 weeks, followed by (iii) single-blind discontinuation for 1 week and open label stimulation for up to 2 years.	Implantation Bilateral Stimulation Monopolar Frequency 130 Hz Amplitude 4–8 mA Pulse Width 91 µs	HRSD-17, BDI-II, GAF.	Reduced depression and increased function. 11 patients responded ** and seven further remitted ^□□^ after 2 year DBS. Efficacy was similar for patients with MDD and those with BP. A modest sham stimulation effect was found, likely due to a decrease in depression after the surgical intervention, but prior to entering the sham phase.	Anxiety; Worsened depression; Nausea; Headache; Infection; Suicide attempts.
Lozano et al. (2012) [66]	Both sexes aged 30–60 years; First MDE before 35 years; HRSD-17 score ≥ 20; GAF < 50.	21	Target SCG Study Design DBS for 12 months, open label.	Implantation Bilateral Stimulation N.A. Frequency 110–140 Hz Amplitude 3.5–7 mA Pulse Width 65–182 µs	HRSD-17, CGI-S.	Improved global functioning and less severe depression. 13 patients responded ***, based on HRSD-17 scores.	Gastrointestinal problems; Skin problem; Suicide; Spasms; Weight gain; Insomnia.
Puigdemont et al. (2012) [67]	18–70 years old with MDE; Resistant to pharmacological treatment and at most, a partial response to ECT; HAMD-17 Score ≥ 18.	8	Target SCG Study Design Intraoperative feedback was provided during surgery for electrode placement. DBS began at 48 h postoperative and ended when each patient had stabilized their response for at least three consecutive visits.	Implantation Bilateral Stimulation Monopolar Frequency 135 Hz Amplitude 3.5–5 V Pulse Width 90 µs	HAMD-17, MADRS, CGI.	Seven patients responded ** and three remitted ^□□^ after 6 month DBS. Five patients responded ** and four remitted ^□□^ after 12 month DBS. Three out of four remitted patients after 12 month DBS had remitted after 3 month DBS.	Suicide ideation; Neck pain; Recurrence; Depression; Cephalalgia.
Kennedy et al. (2011) [18]	Current MDE ≥ 12 months; HRSD-17 score ≥ 20; NR ≥ 4 antidepressant therapies.	20	Target SCG DBS patients were monitored for 3–6 years.	Implantation Bilateral Stimulation Monopolar Frequency 124.7 Hz (average) Amplitude 4.3 V (average) Pulse Width 70.6 µs	HAMD-17 36-item Short-Form Healthy Survey Questionnaire.	64.3% patients responded ** at the last follow-up visit. 35% patients remitted ^□□^ at the last follow-up visit. Scores at the last visit tended towards maintenance of therapeutic scores at 3 years.	Depression; Suicidal thoughts; Suicide (All determined to be unrelated to DBS).
Guinjoan et al. (2010) [68]	Chronic TRD; Family history of affective disorders; Poor response to antidepressants, ECT, and psychotherapy.	1	Target BA25 Study Design Positioning was aided by intraoperative feedback. Bilateral DBS was conducted for 12 months. Followed by unilateral-left, then right DBS, for 6 months	Implantation Bilateral Stimulation Monopolar Frequency 130 Hz Amplitude 3.5–5 V Pulse Width 90 µs	HAMD-17, BDI.	Patient’s condition plateaued after 6 month bilateral DBS. Left unilateral DBS led to rapid worsening in mood. Right unilateral DBS reversed the symptoms and the patient made significant improvements over bilateral stimulation. Patient remitted at 18 months.	Orthostatic hypotension.
Holtzheimer and Mayberg (2010) [69]	Showed signs and symptoms of MDD; Had suicidal ideation in current MDE; Did not improve in symptoms with pharmacotherapy, psychotherapy, and ECT HAM-D score was 25	1	Target SCC Study Design Positioning of electrodes was aided by intraoperative feedback 24 weeks of open-label DBS and chronic stimulation beyond the assessment	Implantation Bilateral Stimulation Monopolar Frequency 130 Hz Amplitude 6 mA Pulse Width 91 µs	HAMD-17	HAMD-17 score lowered to 9 at the end of DBS follow-up. Sustained antidepressant response up to 2 years after surgery	N.A.
Hamani et al. (2009) [70]	Diagnosed MDD; current MDE ≥ 12 months; HAMD-17 score >20; GAF ≤ 50; NR ≥ 4 antidepressant therapies.	20	Target SCG Study Design DBS began at 2 weeks after surgery and lasted for 12 months	Implantation Bilateral Stimulation Monopolar Frequency 130 Hz Amplitude 3–5 V Pulse Width 90 µs	HAMD-17	11 responded ** at the end of 6 month DBS follow-up. Electrodes in responders were positioned ventrally relative to the landmarks of the medial prefrontal lobe.	N.A.
Puigdemont et al. (2009) [71]	Suffered from MDD, with several MDE accompanied by psychotic symptoms; Responded poorly to pharmacotherapy and ECT; Relapse following SCG-DBS with different features; Psychotic as opposed to depressive from previous episodes.	1	Target SCG Study Design DBS for 4 months, then switched off because of relapse and administered ECT for 3 weeks Resumed DBS until 12 months from the beginning of DBS.	Implantation Bilateral Stimulation Monopolar Frequency 135 Hz Amplitude 3.6 V Pulse Width 90 µs	HAMD-17	Sustained response without the need of ECT before relapse. Maintained remission in DBS after ECT until the end of follow-up.	N.A.
Lozano et al. (2008) [72]	Current MDE ≥ 12 months; HRSD-17 score ≥ 20; NR ≥ 4 antidepressant therapies.	20	Target SCG Study Design Blinded-DBS in between and after surgery, monitored for up to 1 year.	Implantation Bilateral Stimulation Monopolar Frequency 130 Hz Amplitude 3.5–5 V Pulse Width 90 µs	HRSD-17, Beck Anxiety Inventory, BDI, CGI-S, PET scans, Neuropsychological tests.	Mean HRSD-17 score lower than baseline at all time points. 12 patients responded ** to DBS, 7 remitted ^□□^ after 6 month DBS. 11 patients responded ** to DBS, 7 were nearly remitted or remitted ^□□^ after 12 month DBS. Eight responses maintained from 6 month to 12 month DBS. PET Scans: decreases in orbital, medial prefrontal cortex, and insula. Increases in lateral prefrontal cortex, parietal, anterior, and posterior cingulate by 6 months; increases in metabolic activity in regions adjacent to SCG.	Seven patients without adverse effects. Wound Infection; Headache; Pain; Seizure; Worsened mood; Irritability.
McNeely et al. (2008) [73]	Current MDE ≥ 12 months, HRSD-17 score ≥ 20, Non-responsive (NR) ≥ 4 antidepressant therapies,	6	Target BA25 Continuous DBS for 12 months.	Implantation Bilateral Stimulation Monopolar Frequency 130 Hz Amplitude 3–4.5 V Pulse Width 60 µs	HRSD-17 Object alternation Test Iowa gambling task Visual delayed recall memory Verbal delayed memory Verbal list learning Stroop color-word	6 months: four responded at the end of DBS ** General Neuropsychological Performance: Manual Motor Skills: Improved for dominant and non-dominant hand by 12 months. Verbal learning: Restored impairments in two patients at the end of 12 months. No significant correlations between change in mood and neuropsychological function at 6 and 12 months.	N.A.
Neimat et al. (2008) [74]	Family history of severe MDD. Failed to respond to antidepressants, adjuncts, and ECT. Relapsed after ablative cingulotomy	1	Target BA25 Study Design Started DBS on the day after electrode implantation and lasted for 30 months	Implantation Bilateral Stimulation Monopolar Frequency 130 Hz Amplitude 4.5 V Pulse Width 60 µs	HAMD-17	HAMD-17 score decreased from 19 before surgery to 8 at 6 months after DBS. Sustained remission until the end of DBS study (scored 7)	N.A.
Mayberg et al. (2005) [19]	Current MDE ≥ 12 months, HRSD-17 score ≥ 20, Non-responsive (NR) ≥ 4 antidepressant therapies.	6	Target BA25 Study Design 1–5 min on-off stimulation in acute DBS for 5 days postoperative. Chronic DBS for 6 months after pulse generator was implanted and optimized for 4 wks	Implantation Bilateral Stimulation Monopolar Acute: Frequency 10–130 Hz Amplitude 0.0–9.0V Pulse Width 30–250 µs Chronic: Frequency 130 Hz Amplitude 4 V Pulse Width 60 µs	HDRS-17, MADRS, CGI, Positive and Negative Affective Scale.	Acute effects: Sudden feeling of calmness Chronic effects: five patients responded ** after 2 month DBS. Response maintained in four patients at the end of 6 month DBS. Three patients achieved remission ^□□^ or near remission at the end of 6 month DBS.	Mild Lightheadedness; Psychomotor slowing; Skin infection; Skin erosion.

* ≥40% reduction in MADRS and average GAF in months 4–6 not worse than baseline; ** ≥50% reduction in HRSD-17 (HAMD-17) score from baseline; *** ≥40% reduction in HRSD-17 score from baseline; ^◊^ ≥40% reduction in MADRS compared to mean baseline; ^□^ HAMD-24 scores or MADRS scores ≤ 10 after DBS; ^□□^ HRSD score < 8. ^□□□^ HRSD score ≤ 8; Abbreviations: ATHF = Anti-depressant Treatment History Form, BA25 = Brodmann Area 25, BDI/-II = Beck Depression Inventory/-II, CGI, PGI, CANTAB = Clinician and Patient Global Impression of Severity and Improvement (CGI; PGI) and cognitive function (CANTAB); CVLT = California verbal learning test, DBS = deep brain stimulation, DWI = Diffusion-weighted imaging, ECG = electrocardiogram, ECT = electroconvulsive therapy, EDA = electrodermal activity, EEG = electroencephalography, GAF = Global assessment function, HAM-A = Hamilton Anxiety Rating Scale, HRSD-17/HDRS-17 = Hamilton Rating Scale for Depression/ Hamilton Depression Rating Scale, (f)MRI = (functional) magnetic resonance imaging; MADRS/MARDS = Montgomery-Åsberg Depression Rating Scale, MAOI = monoamine oxidase inhibitors, MDD = major depressive disorder, MDE = major depressive episodes, MMSE = Mini-Mental State Examination, NSAID = non-steroidal anti-inflammatory drug, NR = non-responsive, PET = positron emission tomography, QIDS/-SR = Quick Inventory of Depressive Symptomatology/-self report, Q-LES-Q = Quality of Life and Satisfaction Questionnaire, SCC = subcallosal cingulate, SCG = subcallosal cingulate gyrus, SCR = skin conductance response, QOL = Quality of Life Enjoyment and Satisfaction Questionnaire, TRD = treatment-resistant depression, WSAS = Work and Social Adjustment Scale.

**Table 2 jcm-09-03260-t002:** Summary of response and remission rates from clinical studies.

Authors	≤ 6 months		6–12 months		12–24 months		≥ 24 months	
	Response (%)	Remission (%)	Response (%)	Remission (%)	Response (%)	Remission (%)	Response (%)	Remission (%)
Sankar et al. 2020 [40]	NA	NA	NA	NA	NA	NA	NA	NA
Riva Posse et al. (2019) [41]	NA	NA	NA	NA	NA	NA	NA	NA
Eitan et al. (2018) [42]	NA	NA	NA	NA	44.4	NA	NA	NA
Merkl et al. (2018) [43]	37.5	12.5	43	14.2	33	33	33	NA
Howell et al. (2018) [44]	-	-	33.3	66.7	-	-	-	-
Waters et al. (2018) [45]	NA	NA	NA	NA	NA	NA	NA	NA
Smart et al. (2018) [46]	-	-	78.5	-	-	-	-	-
Choi et al. (2018) [47]	NA	NA	NA	NA	NA	NA	NA	NA
Conen et al. (2018) [48]	-	-	28.6	42.9	-	-	-	-
Holtzheimer et al. (2017) [15]	22 20 (sham)	10 7 (sham)	28 30 (sham)	12 7 (sham)	54 52 (sham)	17 20 (sham)	48 44 (sham)	25 12 (sham)
McInerney et al. (2017) [14]	-	-	55	-	-	-	-	-
Riva-Posse et al. (2018) [49]	72.7	54.5	81.8	54.5	-	-	-	-
Tsolaki et al. 2017 [50]	50	-	-	-	-	-	-	-
Accolla et al. (2016) [51]	-	-	-	-	79	20	-	-
Richieri et al. (2016) [52]	100 (Case Study)	-	-	-	-	-	-	-
Hilimire et al. (2015) [53]	NA	NA	NA	NA	NA	NA	NA	NA
Martin-Blanco et al. (2015) [54]	NA	NA	NA	NA	NA	NA	NA	NA
Puigdemont et al. (2015) [55]	-	80	-	-	-	-	-	-
Serra-Blasco et al. (2015) [56]	-	-	-	-	75 (F.E.) 87 (TRD)	-	-	-
Choi et al. 2015 [57]	NA	NA	NA	NA	NA	NA	NA	NA
Sun et al. 2015 [58]	NA	NA	NA	NA	NA	NA	NA	NA
Perez-Caballero et al. (2014) [59]	50	-	-	-	-	-	-	-
Merkl et al. (2013) [60]	-	-	-	30	-	-	-	-
Ramasubbu et al. (2013) [61]	50	-	-	-	-	-	-	-
Torres et al. (2013) [62]	100 (2 Case Studies)	-	100 (2 Case Studies)	-	100 (2 Case Studies)	-	100 (2 Case Studies)	-
Broadway et al. (2012) [63]	50	-	-	-	-	-	-	-
Hamani et al. (2012) [64]	-	100 (Case Study)	100 (Case Study)	-	-	-	-	-
Holtzheimer et al. (2012) [65]	18	41	36	36	58	92	-	-
Lozano et al. (2012) [66]	57	-	48	-	62	-	-	-
Puigdemont et al. (2012) [67]	37.5	37.5	87.5	37.5	62.5	50	-	-
Kennedy et al. (2011) [18]	-	-	62.5	-	46.2	-	75	-
Guinjoan et al. (2010) [68]	100 (Case Study)	-	100 (Case Study)	-	100 (Case Study)	-	-	100 (Case Study)
Holtzheimer and Mayberg (2010) [69]	100 (Case Study)	-	100 (Case Study)	-	100 (Case Study)	-	100 (Case Study)	-
Hamani et al. (2009) [70]	-	-	55	-	-	-	-	-
Puigdemont et al. (2009) [71]	100 (Case Study)	-	100 (Case Study)	-	-	-	-	-
Lozano et al. (2008) [72]	35	10	60	35	-	-	-	-
McNeely et al. (2008) [73]	66	NA	-	-	-	-	-	-
Neimat et al. (2008) [74]	100 (Case Study)	NA	-	-	-	-	100 (Case Study)	-
Mayberg et al. (2005) [19]	66	50	-	-	-	-	-	-
**Average**	**63.8%**	**43.9%**	**66.5%**	**36.5%**	**69.3%**	**42.4%**	**76%**	**62.5%**
**Range**	**18–100**	**10–100**	**28.6–100**	**12–66.7**	**33–100**	**17–92**	**33–100**	**12–100**

**Table 3 jcm-09-03260-t003:** A list of preclinical studies on deep brain stimulation of the medial prefrontal cortex in rodents.

Authors	Target	Animal	Animal Models & DBS Design	Stimulation Parameters	Behavioral Tests	Outcomes
Jia et al. (2019) [116]	vmPFC	Sprague-Dawley rats, male.	CUS animal model. Open field test and forced swim test before DBS.	Unipolar High Frequency Frequency: 130 Hz Amplitude: 100 μA Pulse Width: 90 μs Low Frequency Frequency: 20 Hz Amplitude: 400 μA Pulse Width: 0.2 μs	Sucrose preference test	CUS rats had a lowered sucrose preference compared to control rats.
					Open field test	No significant difference in locomotion was recorded between CUS and control groups.
					Forced swim test	Both High- and Low-Frequency Stimulation reduced immobility compared to sham rats.
Papp et al. (2019) [117]	vmPFC	Wistar-Kyoto rats, male (DBS) Wistar rats, male [Venlafaxine(VFX)-treated]	CMS animal model. Two, 2-h DBS sessions were conducted, one on the preceding evening and the other on the following morning before each sucrose intake test and the NORT T1 session.	Frequency: 130 Hz Amplitude: 250 μA Pulse width: 90 μs	Sucrose intake test	During the first 2 wks of CMS, sucrose intake decreased >50% across groups. VFX treatment restored sucrose intake levels.
					Novel object recognition test	Wistar Kyoto rats: DBS rescued novel object recognition test across all groups. Wistar rats: VFX rescued novel object recognition test in CMS animals administered with D2 antagonist, but not in D2-administered CMS animals. VFX also did not rescue groups administered with D3 antagonist.
Bhaskar et al. (2018) [105]	vmPFC	Wistar rats, male.	Naïve animal model. DBS for 15 min prior to and throughout behavioral testing.	Bipolar Frequency: 100 Hz Amplitude: 200 µA Pulse Width: 100 µs	Home-cage emergence test	Enriched environment potentiated the efficacy of HFS on reduced escape latency time in the Naïve animal model.
					Elevated plus maze	HFS with an enriched environment reduced the anxiety index and increased head dips.
					Novel object recognition test	No significant difference.
Bregman et al. (2018) [97]	vmPFC	SERT homozygous knockout and wildtype mice, male.	Serotonin transporter (SERT) knockout model. DBS for 4 h before forced swim test and open field test.	Bilateral Monopolar Frequency: 130 Hz Amplitude: 100 µA Pulse Width: 90 µs	Forced swim test	Both wild-type and knockout-DBS mice had reduced immobility time compared to sham.
					Open field test	Knockout-DBS mice had lower locomotion counts than sham and wild-type mice.
Lehto et al. (2018) [114]	IL	Sprague-Dawley rats, male.	Naïve animal model. All stimulation paradigms consisted of three blocks of 60 s of rest and 18 s of stimulation, ending with an additional rest period, giving a total paradigm of 4 min 54 s.	Monopolar Frequency: 20/35/70/100/130/160/200 Hz tested in randomized order. Amplitude: 1.4–1.7 mA distributed equally among the three electrode channels Pulse Width: 180-μs	N.A.	fMRI conducted to characterize changes in the brain following DBS. IL-DBS at varying stimulation parameters significantly triggered the amygdala. Orientation selective stimulation was able to recruit neuronal pathways of distinct orientations relative to the position of the electrode.
Papp et al. (2018) [118]	vmPFC	Wistar rats, male.	CMS animal model. Two, 2-h DBS sessions were conducted, one on the previous evening and one the next morning 15 min before each behavioral test.	Bipolar Frequency: 130 Hz, Amplitude: 250 μA, Pulse Width: 90 μs	Sucrose intake test	DBS increased sucrose intake across all treatment groups, except for imipramine-treated animals.
					Elevated plus maze	DBS increased the anxiolytic open arm entries in all treatment groups.
					Novel object recognition test	DBS rescued the abolished novel object recognition in CMS sham-treated animals, across all treatment groups.
Perez-Caballero et al. (2018) [115]	IL	Wistar rats, male.	Six independent sets of animals using naïve (unoperated controls) and DBS-off animals.	N.A.	Paw-pressure test	Ibuprofen, tramadol, and morphine significantly increased the paw withdrawal threshold in naïve animals relative to respective vehicle alone, demonstrating a clear analgesic effect.
					Open field test	No analgesics altered the motor activity of rats.
					Modified forced swim test	Electrode implantation induced a significant reduction in the immobility scores of vehicle-treated animals. Ibuprofen abolished the antidepressant-like effect of electrode implantation in the modified forced swim test, increasing the DBS-off animal’s immobility. Neither morphine nor tramadol counteracted the antidepressant-like effect of DBS-off animals.
					Novelty suppressed feeding test	Electrode implantation reduced latency to feed compared to naïve animals. Ibuprofen increased latency to feed relative to VEH-treated animals. Neither morphine nor tramadol reduced the latency to feed in electrode-implanted animals.
Torres-Sanchez et al. (2018) [110]	vmPFC	Wistar rats, male.	Naive animal model. DBS for (i) 4 h at 24 h after surgery, then (ii) 2 h at 48 h after surgery	Bipolar Monophasic Frequency: 130 Hz Amplitude: 100 µA Pulse Width: 90 µs	Forced swim test	Reduced immobility time and increased climbing compared to control.
Volle et al. (2018) [119]	vmPFC	Sprague-Dawley rats, male.	Stimulation was delivered 1 week after surgery for either (i) a single day (acute stimulation; 8 h/day) or (ii) 12 days (chronic stimulation daily for 8 h/day) using a portable stimulator (ANS model 3510) to different groups of rats	Frequency: 130 Hz Amplitude: 200 μA Pulse Width: 90 μs	N.A.	Both treatments increase serotonin (5-HT) release, although fluoxetine resulted in a higher sustained concentration, even upon chronic treatment. Chronic DBS resulted in lowered 5HT release by Day 12. DBS reduced raphe SERT expression. DBS induced changes in 5-HT1B receptor expression, whereas fluoxetine induced changes in 5-HT1A receptors expression in the prefrontal cortex. Research highlighted different effects of both treatments on the serotonergic system.
Reznikov et al. (2017) [109]	IL	Sprague-Dawley rats, male.	Posttraumatic stress disorder animal model. 3-day fear conditioning DBS from 1 week after extinction recall to the end of experiment, 8 h/day, or 2 h before and 4 h after behavioral tests.	Frequency: 130 Hz Amplitude: 100 µA Pulse Width: 90 µs	Extinction recall test	Higher freezing scores in DBS-weak extinction than DBS-strong extinction.
					Open field test	No significant difference between groups.
					Novelty suppressed feeding test	Reduced latency to feeding in DBS-weak extinction, but not strong extinction.
					Elevated plus maze	No significant difference observed between groups.
Bruchim-Samuel et al. (2016) [96]	vmPFC VTA	Flinders Sensitive Line rats, male. Sprague-Dawley rats, male. (Control)	Flinders Sensitive Line model. DBS for 15 min/day, for 10 days.	Bilateral, Monopolar vmPFC Stimulation Frequency: 20 Hz Amplitude: 400 µA Pulse Width: 200 µs VTA Stimulation (control) Frequency: 10 Hz Amplitude: 300 µA Pulse Width: 200 µs	Sweetened condensed milk intake test	No significant difference between vmPFC groups. Significant difference between VTA groups for Flinders Sensitive Line rats.
					Novelty Exploration Test	No significant difference between vmPFC groups. Significant difference between VTA groups for Flinders sensitive line rats.
					Forced swim test	Decreased immobility for vmPFC-stimulated rats after DBS for 10 days, half relapsed at day 28. VTA-stimulated Sprague-Dawley rats had persistently reduced immobility until the end of the experiment.
Jiménez-Sánchez et al. (2016) [33]	IL	Wistar rats, male.	Olfactory bulbectomized model animal model. DBS 1 h daily stimulation, beginning 2 days after electrode implantation before behavioral testing.	Bipolar Biphasic Frequency: 130 Hz Amplitude: 200 µA Pulse Width: 90 µs	Social interaction test	Increased duration of active contact.
					Sucrose preference test	Increased percentage of sucrose consumption in total liquid consumption.
					Forced swim test	Reduced immobility and increased climbing but not swimming.
					Hyperemotionality test	Reduced total behavioral scores when compared to olfactory bulbectomized sham rats.
Jiménez-Sánchez et al. (2016) [81]	IL PrL	Wistar rats, male.	Naïve animal model. DBS for 1 h daily before behavioral testing.	Bipolar Biphasic Frequency: 130 Hz Amplitude: 200 µA Pulse Width: 90 µs	Forced swim test	Reduced immobility and increased climbing in IL-DBS No significant behavioral changes in PrL-DBS.
					Open field test	Insignificant locomotor changes in IL-DBS.
					Novelty suppressed feeding test	Decreased latency to feed in IL-DBS.
Rummel et al. (2016) [113]	vmPFC	Flinders Sensitive Line rats, male. Congenitally learned helplessness rats, Male.	Experiment 1 Chronic intermittent DBS 1 week after surgery in Flinders sensitive line rats, 30 min each morning and extra 30-min stimulation on afternoons before the day of behavioral test. Experiment 2 Chronic continuous DBS 1 week after surgery for 16 days. Experiment 3 Chronic intermittent DBS in congenitally learned helpless rats, procedures followed that in experiment 1.	Chronic intermittent DBS Frequency: 130 Hz Amplitude: 300 µA Pulse Width: 100 µs Chronic continuous DBS Frequency: 130 Hz Amplitude: 150 µA Pulse Width: 100 µs	Sucrose consumption test	Chronic intermittent DBS increased sucrose intake in Flinders sensitive line rats but not in congenitally learned helplessness rats. Chronic continuous DBS did not affect sucrose intake in Flinders sensitive line rats and congenitally learned helplessness rats.
					Forced swim test	Chronic intermittent DBS and chronic continuous DBS increased latency to immobility in Flinders sensitive line rats but not congenitally learned helplessness rats.
					Learned helplessness paradigm	Chronic intermittent DBS and chronic continuous DBS decreased helplessness in Flinders sensitive line rats but not cLH rats.
					Sucrose intake test	No significant difference observed.
					Novelty exploration test	No significant difference observed.
Bambico et al. (2015) [98]	vmPFC	Fisher rats, Male.	CUS animal model CUS for ~4 weeks until anhedonia inferred by SPI scores, then performed implantation DBS for 3 weeks after implantation, 8 h per day, 7 days per week	Frequency: 130 Hz Amplitude: 100 µA Pulse Width: 90 µs	Novelty-suppressed feeding test	Reduced latency to feeding in CUS-DBS animals compared to CUS-sham animals.
					Open field test	No significant difference observed.
					Elevated plus maze test	More time in open arms in CUS-DBS animals compared to CUS-sham animals.
					Forced swim test	Reduced immobility time in CUS-DBS animals compared to CUS-sham animals.
Edemann-Callesen et al. (2015) [94]	vmPFC; Medial forebrain bundle.	Flinders sensitive line rats, male. Sprague-Dawley rats, male.	Naïve animal model DBS was applied in an Intra-cranial self- stimulation protocol.	Bilateral, Monopolar Frequency: 20–200 Hz Amplitude: 170–560 µA Pulse Width: 100 µs	Intra-cranial self-stimulation	For Flinders Sensitive Line rats, vmPFC-DBS did not affect the reward-seeking behavior compared to medial forebrain bundle DBS.
Etiévant et al. (2015) [103]	IL	Sprague-Dawley rats, male.	Naïve animal model. DBS for 4 h after forced swim pre-test and 2 h before forced swim test.	Bipolar Unilateral Frequency: 130 Hz Amplitude: 150 µA Pulse Width: 60 µs	Forced swim test	Reduced immobility duration in IL-DBS compared to control.
Insel et al. (2015) [111]	IL	Sprague-Dawley rats, male.	Naïve animal model. DBS for 8 h per day, for 10 days.	Monopolar Bilateral Frequency: 130 Hz Amplitude: 100 μA Pulse Width: 90 μs	Spontaneous behavior recording	Coherence between ventral hippocampus and IL was reduced after 10-day DBS compared to sham in 2–4 Hz brain activity range, but was not reduced after only 1 day of treatment. Coherence was not affected by fluoxetine, indicating that IL-DBS observations were independent of the serotonergic pathways.
Lim et al. (2015) [102]	vmPFC	Sprague-Dawley rats, male.	Experiment 1 Naïve animal model. Experiment 2 CUS animal model. CUS for 3 weeks, each stressor lasted for 10–14 h DBS for 15 min before home-cage emergence test, before and during open field test.	Bipolar Biphasic Experiment 1 Frequency:10/100 Hz Amplitude: 100 µA Pulse Width: 100 µs Experiment 2 Frequency: 100 Hz Amplitude: 100µA Pulse Width: 100µs	Home-cage emergence test	HFS reduced escape latency time in Naïve and CUS animal model.
					Open-field test	Insignificant effect in Naïve animals for both HFS and LFS. Increased time spent in the central zone for HFS-CUS.
					Food-intake test	HFS increased food intake in naïve animals. No significance difference observed in CUS animals.
					Sucrose-intake test	Insignificant in Naïve animals for both HFS and LFS. Increased sucrose intake in HFS-CUS.
					Forced swim test	Insignificant in Naïve for both HFS and LFS. Reduced immobility duration in HFS-CUS.
Lim et al. (2015) [108]	PrL	Sprague-Dawley rats, male.	Naïve animal model. DBS for 15 min before and during sucrose intake test (same for forced swim test) and before sacrifice for 1 h.	Bipolar Biphasic Frequency: 100 Hz Amplitude: 100 µA Pulse Width: 100 µs	Sucrose intake test	Increased sucrose consumption.
					Forced swim test	Reduced immobility.
Liu et al. (2015) [99]	vmPFC	Sprague-Dawley rats, male. 4 months old and 12 months old.	Acute DBS Naïve animal model. DBS for 30 min before the behavioral tests. Chronic DBS Naïve animal model. DBS for 1 h daily including days of behavioral tests.	Bipolar Biphasic Acute DBS Frequency: 10/100 Hz Amplitude: 50/100/200/400 µA Pulse Width: 100 µs Chronic DBS Frequency: 100 Hz Amplitude: 100 µA Pulse Width: 100 µs	Novel object recognition test	Acute HFS at 200 µA produced higher novel object exploration than familiar object in short-term memory test. Chronic HFS increased novel object exploration in short- and long-term memory than familiar object, as well as the durations.
					Morris water maze	Shorter latency to reach platform on day 1 and 2 in chronic HFS compared to sham. More time spent in the target quadrant and less in the opposite quadrant in chronic HFS compared to sham.
Hamani et al. (2014) [100]	vmPFC	Sprague-Dawley rats, male.	Naïve animal model. DBS for 4 h after FST on day 1, 2 h before swimming on day 2 DBS 1 week after forced swim test, 4 h on day 1, 2 h on day 2, then assessed in open field test	Monopolar Frequency: 130 Hz Amplitude: 100 µA Pulse Width: 90 µs	Forced swim test	Reduced immobility and increased climbing frequency between groups.
					Open field test	No significant difference in locomotion observed.
Laver et al. (2014) [112]	vmPFC	Sprague-Dawley rats, male.	Naïve animal model. Serotonin reuptake inhibitors/vehicle were injected i.p. 1 h and 5 h after forced swim test on day 1 and 1 h before forced swim test on day 2. DBS for 4 h on day 1 of forced swim pre-test and 2 h before forced swim test on day 2.	Monopolar Bilateral Frequency: 130 Hz Amplitude: 100 μA Pulse Width: 90 μs	Forced swim test	DBS-saline, DBS-buspirone, DBS-Risperidone, DBS-pindolol-treated animals had higher swimming and lower observed immobility frequencies.
					Open field test	No significant difference observed
Perez-Caballero et al. (2013) [45]	IL	Wistar rats, male.	CUS animal model. Electrode implantation after week 4 of CUS, then CUS resumed. DBS for 4 h after forced swim pre-test and 2 h before forced swim test. Some animals received pre-treatment with indomethacin or ibuprofen.	Bipolar Monophasic Frequency: 130 Hz Amplitude: 100 µA Pulse Width: 90 µs	Forced swim test	Reduced immobility and increased swimming in DBS-off-IL and DBS-on-IL compared to sham- and naïve-animals. Increased immobility and reduced swimming in DBS-off-IL animals treated with NSAIDs.
					Open field test	No significant difference.
Rea et al. (2014) [95]	vmPFC	Flinders sensitive line rats, male. Flinders resistant line rats, male. (Control)	DBS for 30 min each morning for 2 weeks. Extra DBS for 30 min in the afternoon before the day of behavioral tests and during behavioral tests.	Monopolar Frequency: 130 Hz Amplitude: 300 µA Pulse Width: 100 µs	Forced swim test	DBS reduced immobility in both groups of rats.
					Sucrose consumption test	DBS increased sucrose consumption in both groups of rats.
Veerakumar et al. (2014) [104]	vmPFC	C57BL/6 mice, male.	Chronic social defeat stress animal model. DBS for 5 h/day, for 7 days.	Bipolar Unilateral Frequency: 160 Hz Amplitude: 150 µA Pulse Width: 60 µs	Social interaction test	Before DBS, defeat-susceptible mice showed lower interaction times. Defeated animals with DBS spent longer in the interaction zone than sham and similar to non-stressed animals. DBS increased the total distance traveled.
Hamani et al. (2012) [92]	vmPFC	Wistar rats, male.	CUS animal model. DBS for 8 h/day, for 2 weeks	Monopolar Frequency: 130 Hz Amplitude: 200 µA Pulse Width: 90 µs	Sucrose preference test	Higher preference observed in CUS-treated DBS animals when compared to CUS-sham animals. Higher sucrose consumption in CUS-treated DBS than CUS-sham alone.
Hamani et al. (2010) [93]	vmPFC, IL, PrL	Sprague-Dawley rats, male	Naïve animal model DBS for 4 h after FST on day 1, 2 h before swimming on day 2	Frequency: 20 Hz/130 Hz Amplitude: 100/200/300/400 µA Pulse Width: 90 µs	Forced swim test	Parameters of 130 Hz, 90 µs, 200 µA reduced immobility the most in vmPFC-DBS, also at 100 µA and 300 µA. PrL stimulation at 130 Hz, 90 µs, 200 µA reduced immobility, but IL stimulation was insignificant.
Hamani et al. (2010) [91]	vmPFC	Sprague-Dawley rats, male	Naïve animal model, with serotonergic depletion, or norepinephrine lesion. DBS for Forced Swim Test Day 1: 4 h after forced swim test Day 2: 2 h before forced swim test DBS for open field test, novelty suppressed feeding test, and learned helplessness. Pre-test Day 1: 4 h Pre-test Day 2: 2 h	Monopolar Frequency: 130 Hz Amplitude: 100 µA Pulse Width: 90 µs	Forced swim test	DBS reduced immobility and increased swimming counts in naïve animals. DBS in animals with ibotenic acid injection had lower immobility and higher swimming counts than control. Rats with DBS without serotonergic depletion exhibited lower immobility than DBS animals with serotonergic depletion. Rats with DBS without norepinephrine lesion had lower immobility than control, similar reduction in immobility was shown in animals with DBS and norepinephrine lesion.
					Open field test	No significant difference.
					Novelty suppressed feeding test	DBS reduced latency to feed compared to control.
					Learned helplessness paradigm	Insignificant difference in escape latency between DBS and control.
			Animals predisposed to helplessness. DBS for 2 h before baseline assessment, 2 h before footshock at 2 days after baseline test, and DBS for 2 h before sucrose consumption test on the next day	Monopolar Frequency: 130 Hz Amplitude: 100 µA Pulse Width: 90 µs	Sucrose consumption test	DBS reduced the sucrose drinking time in animals after footshock, but this was insignificant.

Abbreviations: Cg, cingulate cortex; CUS, chronic unpredictable stress; EPM, elevated plus maze; FST, forced swim test; HFS, high-frequency stimulation; IL, infralimbic cortex; LFS, low-frequency stimulation; MWM, Morris water maze test; NORT, novel object recognition test; NSFT, novelty-suppressed feeding test; OBX, olfactory bulbectomy; OFT, open field test; PrL, prelimbic cortex; SD, Sprague-Dawley; SPI, sucrose preference index; vmPFC, ventromedial prefrontal cortex.

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
