# Peer review of "A Decade of Progress in Deep Brain Stimulation of the Subcallosal Cingulate for the Treatment of Depression"

_jcm, 2020, doi:10.3390/jcm9103260_

Round 1

Reviewer 1 Report

The authors have address all my concerns. 

Author Response

Dear Reviewer,

Thank you for your constructive feedback, which has improved the quality of our review.

Reviewer 2 Report

Khairuddin et al present a review article in which they aim to provide information on the progress in the use of DBS for the treatment of refractory depression. The manuscript is long and its organisation is not optimal, with data on human studies appearing before data on the rodent studies that contributed to the development of human trials. The writing is somewhat disjointed and the flow of the article in terms of chronology and  general organisation is poor. The paper is largely composed of disjointed sentences referring to the findings of  other papers without any effort to provide a meaningful interpretation and analysis of the topic that might be accessible to the reader. 

I have some major concerns about the use of references. Take for example line 42:

'However, newer generations of antidepressants 43 were barely more effective than first-generation tricyclic antidepressants [10]'  

If the paper by McInerney et al reference [10] refers to is read, in the first paragraph the statement appears:

'Despite advances in drug development to treat major depressive disorder (MDD), there is no evidence that newer drugs have higher efficacy compared to first generation tricyclic antidepressants (Baghai et al., 2011)'

So rather than directly referencing the original seminal paper by Baghai, a second-hand reference to it is provided, by quoting inaccurately a line from a different paper using the same reference. This trend is continues throughout the paper and is not acceptable in my opinion as it raises a question as to whether the relevant seminal papers are being assessed.

Line 51 states that DBS was first used to treat 'movement disorders in patients with Parkinson's disease [sic]'. This is not accurate. DBS was first appleid to the treatment of essential tremor in Grenoble in 1987. Again, the references are totally inappropriate, instead of referencing the seminal papers from Grenoble, other papers that presumably use the references appropriately are referenced. This approach to references is very frustrating.

In the background the authors state 'a broad-acting safe therapy needs to be developed' before going on to discuss developments in DBS and depression. They fail however to demonstrate that DBS is either broad acting or safe, with a particular lack of information on safety data, inparticular around suicide and stimulation related side effects. Lines 185-197 are provided for a summary of adverse events in which the authors list off individual complications from selected papers. A more comprehensive review of all common or major device and stimulation related complications (infection, haemorrhage, lead misplacement) with a focus on suicide risk from DBS studies on refractory depression - all targets - would be informative and important in this patient group). 

Table 1 is excessively long and table 2 is largely redundant with most data fields being empty. 

The paragraph on other targets for treatment of depression (Line 315) is far too short and omits important sham controlled trials by Raymaekers et al (inferior thalamic peduncle and internal capsule) and Dougherty et al (internal capsule). A further recent sham controlled trial of DBS to the med forebrain bundle by Coenen et al is also not referenced.

The conclusion is short for a paper of this length and weak in its writing. The reader is left with no clear impression of the state of the science or technology despite a title which promises to reveal the progression over the previous decade. 

Author Response

Dear Sir/Ms., 

Round 2

Reviewer 2 Report

Khairuddin an colleagues submit a revised manuscript. While they sand firm on some issues, providing rationale and reasoned explanation, they have made efforts to amend oter areas of concern. In particular they have addressed, as far as I can see, the issue around appropriate citations.

I have no additional comments to make and wish the authors well. The manuscript is acceptable in its current form.